DOI: 10.1038/s41467-018-03243-7　　**OPEN**

# Clonally diverse CD38$^+$HLA-DR$^+$CD8$^+$ T cells persist during fatal H7N9 disease

Zhongfang Wang[1,2], Lingyan Zhu[1], Thi H.O. Nguyen[2], Yanmin Wan[1], Sneha Sant[2], Sergio M. Quiñones-Parra[2], Jeremy Chase Crawford [3], Auda A. Eltahla[4], Simone Rizzetto[4], Rowena A. Bull[4], Chenli Qiu[1], Marios Koutsakos[2], E. Bridie Clemens[2], Liyen Loh[2], Tianyue Chen[1], Lu Liu[1], Pengxing Cao[5], Yanqin Ren[1], Lukasz Kedzierski[2], Tom Kotsimbos[6], James M. McCaw [5], Nicole L. La Gruta[2,7], Stephen J. Turner[2,8], Allen C. Cheng[9], Fabio Luciani[4], Xiaoyan Zhang[1], Peter C. Doherty[2,3], Paul G. Thomas[3], Jianqing Xu [1] & Katherine Kedzierska[1,2]

Severe influenza A virus (IAV) infection is associated with immune dysfunction. Here, we show circulating CD8$^+$ T-cell profiles from patients hospitalized with avian H7N9, seasonal IAV, and influenza vaccinees. Patient survival reflects an early, transient prevalence of highly activated CD38$^+$HLA-DR$^+$PD-1$^+$ CD8$^+$ T cells, whereas the prolonged persistence of this set is found in ultimately fatal cases. Single-cell T cell receptor (TCR)-αβ analyses of activated CD38$^+$HLA-DR$^+$CD8$^+$ T cells show similar TCRαβ diversity but differential clonal expansion kinetics in surviving and fatal H7N9 patients. Delayed clonal expansion associated with an early dichotomy at a transcriptome level (as detected by single-cell RNAseq) is found in CD38$^+$HLA-DR$^+$CD8$^+$ T cells from patients who succumbed to the disease, suggesting a divergent differentiation pathway of CD38$^+$HLA-DR$^+$CD8$^+$ T cells from the outset during fatal disease. Our study proposes that effective expansion of cross-reactive influenza-specific TCRαβ clonotypes with appropriate transcriptome signatures is needed for early protection against severe influenza disease.

[1] Shanghai Public Health Clinical Center & Institutes of Biomedical Sciences, Key Laboratory of Medical Molecular Virology of Ministry of Education/Health, Shanghai Medical College, Fudan University, 201508 Shangai, China. [2] Department of Microbiology and Immunology, University of Melbourne, The Peter Doherty Institute for Infection and Immunity, Parkville, Melbourne, VIC 3010, Australia. [3] Department of Immunology, St. Jude Children's Research Hospital, Memphis, TN 38105, USA. [4] School of Medical Sciences and The Kirby Institute, UNSW Sydney, Sydney, NSW 2052, Australia. [5] School of Mathematics and Statistics, University of Melbourne, Parkville, VIC 3010, Australia. [6] Department of Respiratory Medicine Alfred Hospital Health and Department Medicine, Monash University, 99 Commercial Road, Melbourne, VIC 3004, Australia. [7] Department of Biochemistry and Molecular Biology, Infection and Immunity Program, Biomedicine Discovery Institute, Monash University, Clayton, VIC 3800, Australia. [8] Department of Microbiology, Biomedical Discovery Institute, Monash University, Clayton, VIC 3800, Australia. [9] Infection Prevention and Healthcare Epidemiology Unit Alfred Health and School of Public Health and Preventive Medicine, 99 Commercial Road, Melbourne, VIC 3004, Australia. Zhongfang Wang and Lingyan Zhu contributed equally to this work. Jianqing Xu and Katherine Kedzierska jointly supervised this work. Correspondence and requests for materials should be addressed to J.X. (email: xujianqing@shphc.org.cn) or to K.K. (email: kkedz@unimelb.edu.au)

Annual influenza epidemics lead to severe illness, life-threatening complications, and death, especially in high-risk groups such as young children, elderly, pregnant women, obese, individuals with comorbidities and indigenous populations. Disease morbidity and mortality increase when a new influenza strain reasserts or jumps the host, and becomes capable of infecting humans. In this case, there is no (or minimal) pre-existing antibody-mediated immunity to the new viral strain at the population level, leading to millions of infections and a rapid global spread of the virus. In the absence of antibodies, the severity of the disease can be ameliorated by broadly cross-reactive cellular immunity, especially cytotoxic CD8$^+$ T cells[1–5]. However, the precise mechanism of how CD8$^+$ T cells mediate recovery in some individuals, but not others, is far from clear.

The novel, avian-origin triple-reassortant A/H7N9 influenza A virus (IAV) that emerged in China during 2013[6] causes severe human disease, with >99% hospitalization rates, 75% ICU admissions, >71% acute respiratory distress syndrome, and ~40% mortality. From October 2016, the H7N9 "fifth wave" has been responsible for 713 known human cases and 205 deaths. New mutations within the haemagglutinin (HA) cleavage sites of H7N9 have raised concerns regarding adaption for human-to-human transmission and, though this is yet to occur, H7N9 is (along with other pathogenic IAVs) a potential pandemic threat. Longitudinal analysis of immune response dynamics in a unique cohort of hospitalized H7N9 patients at the Shanghai Public Health Clinical Center (SHAPHC)[5,7] associated early recovery with the generation of robust IFN-γ-producing CD8$^+$ T-cell populations soon after admission. Conversely, delayed emergence of this population associated with an increased prevalence of CD4$^+$ T cells and NK cells was observed in patients with longer hospital stays[5]. Fatal outcomes were associated with minimal evidence of IAV-specific immunity and diminished T-cell function at the cellular and transcriptome levels. H7N9, together with other avian influenza viruses, constitutes a potential pandemic threat; as such it is important to understand the key differences in human immune responses between patients who recover and those whom succumb to fatal influenza disease. The central question here was whether dysfunctional T cells in fatal cases resulted from a total lack of activation consequent to immunosuppression.

Here, we analyzed the activation and recruitment of H7N9-specific CD8$^+$ T cells in survival versus fatal patient groups in a unique longitudinal cohort of samples from the first wave of H7N9 epidemic in China. We hypothesize that lethal H7N9 disease would be associated with defective T-cell activation and a lack of relevant T-cell receptor (TCR) specificities. CD8$^+$ T cells were non-functional (by IFNγ production) in those who succumbed; these CD8$^+$ T cells displayed high and persistent expression of the CD38$^+$HLA-DR$^{+[8–10]}$ activation markers, along with prolonged expression of the inhibitory PD-1 immune checkpoint receptor. Further analysis of TCRαβ clonotypes within A2-M1$_{58}$ tetramer$^+$ and CD38$^+$HLA-DR$^+$CD8$^+$ cells established that, while TCRαβ diversity was similar within single-specificity A2-M1$_{58}$$^+$CD8$^+$ T cells and activated CD38$^+$HLA-DR$^+$CD8$^+$ T cells, TCRαβ repertoires within CD38$^+$HLA-DR$^+$CD8$^+$ T cells utilized a significantly broader range of TCRα and TCRβ gene segments. Interestingly, delayed clonal expansion kinetics and a divergent differentiation pathway associated with the fatal H7N9 patients by our single-cell TCRαβ and RNA sequencing analyses was shown in patients who succumbed to the avian H7N9 influenza viral infection. Overall, our analysis supports the concept that cross-reactive memory TCRαβ$^+$CD8$^+$ T cells capable of large clonal expansions mediate significant early protection against severe influenza disease caused by newly emerging viruses, while prolonged persistence of clonally diverse

CD38$^+$HLA-DR$^+$CD8$^+$ T cells may be associated with poorer clinical outcomes for severe IAV infections.

## Results

**Prolonged CD38$^+$HLA-DR$^+$ expression predicts fatal outcomes.** The analysis utilized a previously studied[5,7] longitudinally obtained PBMC cohort from 11 H7N9-infected patients: eight of whom recovered (a9, a10, a11, a12, a130, a20, a78 and a79) while three died (a22, a33 and a131). Patients discharged within 2–3 weeks had an early peak in H7N9-specific CD8$^+$ T-cell responses based on ex vivo IFNγ production, while those who succumbed secreted minimal IFNγ after H7N9 stimulation[5] (Fig. 1a, d). What mechanisms underlie this perturbed CD8$^+$ T-cell function in fatal cases? We first analyzed the expression of CD38 and HLA-DR, key markers of CD8$^+$ T-cell activation during viral infections[8–10]. As expected, CD38$^+$HLA-DR$^+$ expression on CD8$^+$ T cells in the recovery group was high, although transient (Fig. 1a, b). Early in the infection (~d10), the frequency of CD38$^+$HLA-DR$^+$CD8$^+$ T cells in the recovery group reached a mean of 26.6% (CI 95%: 15.4–37.8% at the earliest time-point), then gradually declined (~d14–d18) to 10.3% (CI 95%: 3.8–16.8%) as patients recovered (Fig. 1a, b). Conversely, patients with fatal disease outcomes displayed high and prolonged expression of CD38$^+$HLA-DR$^+$ on CD8$^+$ T cells, starting with a mean of 18.7% (CI 95%: 2.3–35.1%) and reaching 37.5% (CI 95%: 21.0–54.18%) at the latest time-point, likely reflecting prolonged virus exposure[7]. Thus, the average proportion of activated CD38$^+$HLA-DR$^+$CD8$^+$ T cells, as shown mean values of all the assayed time-points (Fig. 1e), was significantly higher ($p = 0.026$, Wilcoxon rank-sum test) for patients who died (mean = 32.47%) than in the survivors (mean = 18.62%; across a mean hospital stay of 27 days).

Fatal H7N9 infection is characterized by lymphopenia[5] indicating, perhaps, recruitment to the draining lymph nodes and the infected lung. Indeed, calculating the absolute numbers of CD38$^+$HLA-DR$^+$CD8$^+$ T cells per 1 ml of blood (Fig. 1c) shows that, reflecting the lower total CD8$^+$ T-cell counts from fatal patients[5,11], there were more numbers of circulating CD38$^+$HLA-DR$^+$CD8$^+$ T cells (data from all the assayed time-points) in 1 ml of blood in those who recovered ($p = 0.003$, standard two-tail Student's $t$-test; Fig. 1f). Similarly, there were more absolute numbers of H7N9-specific IFNγ$^+$CD8$^+$ T cells in 1 ml of blood in the recovery group, as compared to the fatal group ($p < 0.0001$, standard two-tail Student's $t$-test; Fig. 1g).

Thus, while the frequency of circulating CD38$^+$HLA-DR$^+$CD8$^+$ differs between the surviving and fatal patients, this is not true for CD38$^+$HLA-DR$^+$CD8$^+$ T-cell numbers in 1 ml of blood, analyzed as peak values of all the assayed time-points (Fig. 1h; $p = 0.279$; Wilcoxon rank-sum test). Conversely, the average numbers of IFNγ$^+$CD8$^+$ T cells in 1 ml of blood, analyzed as peak values of all the assayed time-points, in the recovery group were significantly higher than those in the fatal group (Fig. 1i; $p = 0.012$, Wilcoxon rank-sum test), reflecting the minimal influenza-specific CD8$^+$ T-cell response in fatal cases.

Overall, our data show that, while the higher prevalence of IFNγ$^+$CD8$^+$ T cells is indicative of recovery, the proportion of the CD38$^+$HLA-DR$^+$CD8$^+$ set within the total CD8$^+$ T-cell population is enriched over a prolonged period, in PBMC populations from the lymphopenic patients who died.

**CD38$^+$HLA-DR$^+$PD-1$^+$CD8$^+$ prevail in fatal H7N9 cases.** Given that the expression of exhaustion markers (including the negative checkpoint PD-1 molecule[12–14]) is associated with decreased IFNγ production, we analyzed PD-1 expression on activated CD38$^+$HLA-DR$^+$CD8$^+$ T cells from the hospitalized

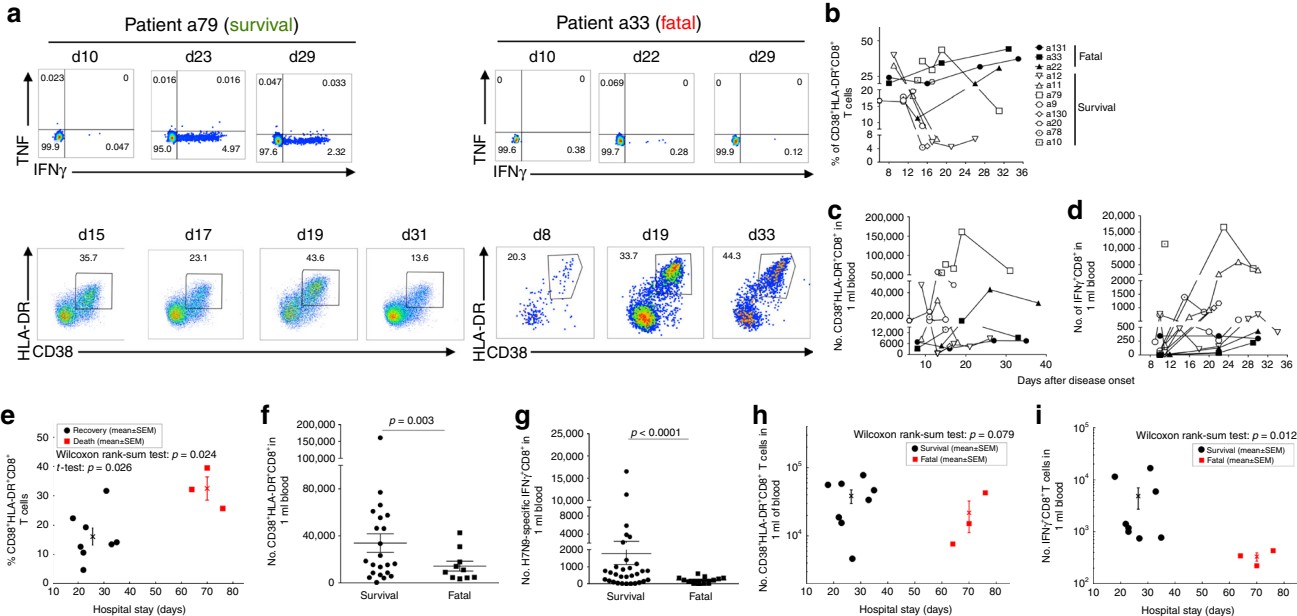

**Fig. 1** Prolonged expression of CD38+HLA-DR+ on CD8+ T cells in fatal H7N9 cases. The frequency of CD38+HLA-DR+CD8+ T cells and IFNγ/TNF production was assessed in the peripheral blood of patients hospitalized with severe H7N9 infection at different times after infection. **a** Representative FACS plots are shown for IFNγ/TNF production, cell gating shown in the previous report[5], and CD38+HLA-DR+ expression by CD8+ T cells for patient a79 (recovered from H7N9) and patient a33 (died from H7N9). The gating strategy is shown in Supplementary Figure 7. **b–i** Data are graphed for all the patients available for this study: eight "recovery" patients (open symbols) and three patients with fatal disease outcomes (solid symbols) as **b** frequency of CD38+HLA-DR+ expression in CD8+ T cells; **c** numbers of CD38+HLA-DR+ CD8+ T cells in 1 ml of blood; **d** numbers of IFNγ+CD8+ T cells in 1 ml of blood; **e** frequencies of activated CD38+HLA-DR+CD8+ T cells, as mean values of all the assayed time-points, are shown for patients who survived versus patients who died; **f** numbers of CD38+HLA-DR+CD8+ T cells (data from all the assayed time-points) in 1 ml of blood were analyzed according to the disease outcome; **g** numbers of IFNγ+CD8+ T cells in 1 ml of blood (data from all the assayed time-points) were analyzed according to the disease outcome; **h, i** binary comparisons between the survival and fatal groups were performed for the hospital stay time (in days). **h** Numbers of activated CD38+HLA-DR+CD8+ T cells in 1 ml of blood, as peak values of all the assayed time-points, are shown for patients who survived versus patients who died; and **i** numbers of activated IFNγ+CD8+ T cells in 1 ml of blood, as peak values of all the assayed time-points, are shown for patients who survived versus patients who died. Analyses were implemented using MATLAB (version R2014b; the MathWorks, Natick, MA). Error bars represent standard errors of the mean. Standard two-tail Student's t-test was performed with p value as indicated

patients. The majority of CD38+HLA-DR+CD8+ T cells were also PD-1+ (Fig. 2a–d) and there was a significant correlation between the CD38+HLA-DR+ and CD38+PD-1+ CD8+ T-cell sets (Fig. 2b; p < 0.001 by Pearson/Spearman tests). Most strikingly, the frequency dynamics of CD8+ T cells expressing all three CD38+HLA-DR+PD-1+ markers again segregated the patients into those who survived and died, accurately predicting divergent disease outcomes (Fig. 2d). Such co-expression of CD38+HLA-DR+ with PD-1 may also explain impaired functionality in terms of IFNγ production.

Interestingly, PD-1 cell surface concentrations, as measured by mean fluorescence intensity (MFI) of PD-1 staining within the CD38+HLA-DR+CD8+ set, varied across the two patient groups (Fig. 2d). While those who recovered had uniform, intermediate levels of PD-1, those with the fatal H7N9 outcomes displayed two peaks of PD-1 staining, representing intermediate and high PD-1 expression (Fig. 2c). Such differential PD-1 profiles have (in mice)[14] been considered to define functional (intermediate PD-1) versus exhausted (high PD-1) CD8+ T-cell populations. Furthermore, as expected, the CD38+PD-1+CD8+ T cells displayed a more activated CD45RA−CD27lo or CD45RA−CD27hi phenotype, in contrast to the total CD8+ T cells found across four quadrants marked by CD45RA and CD27 (Fig. 2e).

**Similar CD38+PD-1+ profiles in seasonal IAV infections**. To understand whether such clinical outcome related to CD38+ PD-1+ CD8+ T-cell expression profiles is also characteristic of

seasonal influenza epidemics, we analyzed (Fig. 2g) CD38+PD-1+ expression on CD8+ T cells for longitudinal PBMC samples from 22 patients hospitalized with PCR-confirmed seasonal influenza and five control hospitalized virus-negative patients. Peripheral blood was taken at hospitalization, 3–5 days later, at discharge and ~30 days after symptom onset. Similar to surviving H7N9 patients (Fig. 2f), patients hospitalized with seasonal influenza (A/H1N1, A/H3N2, and B strains) had high CD38+PD-1+ expression early after infection, followed by a decrease with recovery (p = 0.04, standard two-tail Student's t-test ; Fig. 2i). In contrast, one patient suffering underlying acute myeloid leukemia who died during H3N2 infection displayed prolonged and increasing frequency of CD38+PD-1+ expression on CD8+ T cells (Fig. 2g), comparable to that characteristic of the fatal H7N9 cases (Fig. 2f). Conversely, CD38+PD-1+ expression remained low in patients who were hospitalized with no evidence of any virus infection (Fig. 2g) or, in most instances, following immunization with either trivalent (TIV) or quadrivalent (QIV) split influenza vaccine (Fig. 2h, i). High CD38+PD-1+ expression was characteristic of one vaccinated individual prior to and early after vaccination, suggesting that the individual had some unidentified infection (Fig. 2h). Thus, patterns of CD38+PD-1+ expression on CD8+ T cells during seasonal pH1N1, H3N2, and influenza B virus-induced disease replicated the situation described for the hospitalized H7N9 patients.

Overall, although the CD38+HLA-DR+ phenotype is known to be associated with T cell activation[8–10], here we provide the first evidence that prolonged and dysregulated kinetics of CD38

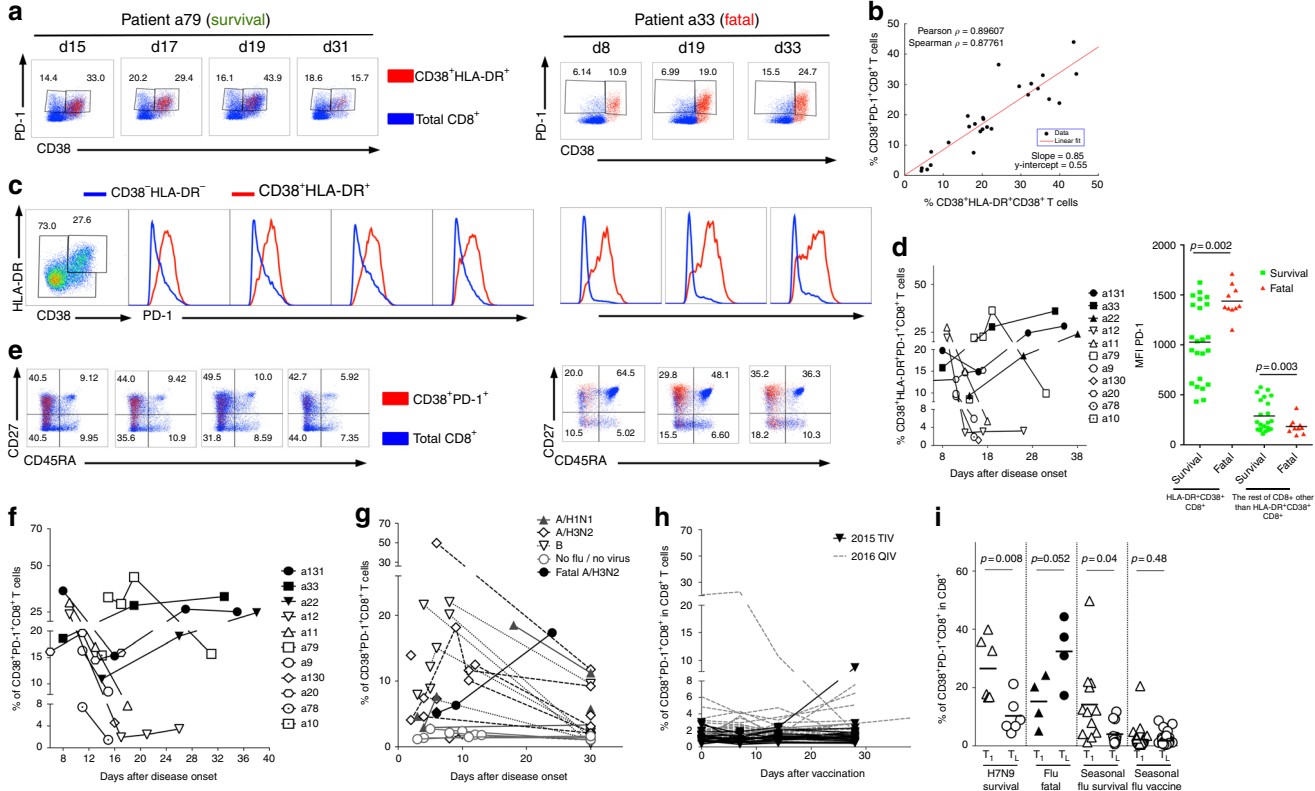

**Fig. 2** Dynamics of CD38+HLA-DR+PD-1+CD8+ cells segregate H7N9 disease outcome. **a** Representative FACS plots of PD-1 expression within the CD38+HLA-DR+CD8+ T cells (in red) and the total CD8+ T cells (in blue) in patient a79 (recovered from H7N9) and patient a33 (died from H7N9). **b** Correlation between the frequency of CD38+PD-1+CD8+ T cells and a proportion of CD38+HLA-DR+CD8+ T cells, as measured by Spearman test. **c** Representative histograms of PD-1 expression within CD38+HLA-DR+CD8+ T cells (red) and the remaining non-CD38+HLA-DR+CD8+ T cells (blue). **d** High and prolonged expression of PD-1 in patients who died, as triple-positive CD38+HLA-DR+PD-1+ and mean fluorescence intensity (MFI) of PD-1 expression within CD38+HLA-DR+CD8+ T cells and non-CD38+HLA-DR+CD8+ T cells for survivals and fatal cases (a standard two-tail Student's t-test), with samples from individual patients being acquired on an LSR Fortessa on different days. **e** CD45RA and CD27 profiles on CD38+PD-1+CD8+ T cells (in red) and on total CD8+ T cells (in blue). **f** Similar to CD38+HLA-DR+CD8+ T cells, the frequency of CD38+PD-1+CD8+ T cells declined with time in the H7N9 survival group, which was maintained or increased in H7N9 patients who died. The frequency of CD38+PD-1+CD8+ T cells was also analyzed in **g** patients hospitalized with seasonal pH1N1 (5 patients; dotted lines, solid triangles), H3N2 (10 patients in dashed lines, open diamonds), influenza B (6 patients in dotted black lines, open triangles), non-influenza/non-viral hospitalized patients (5 patients; gray lines, open gray circles), and 1 H3N2 patient with fatal outcome (black line, solid black circle); and **h** healthy individuals during inactivated influenza vaccination in 2015 (18 individuals vaccinated with a trivalent (TIV) vaccine; black lines) and 2016 (27 individuals vaccinated with quadrivalent (QIV) vaccine; gray lines). **i** Frequency of CD38+PD-1+CD8+ T cells during the first (T₁) and last time-point (T_L) of hospitalization or vaccination in H7N9 survival cases, fatal influenza patients (3 H7N9, 1 H3N2); seasonal influenza survival cases, and seasonal influenza vaccinees. Means are shown as horizontal bars. Standard two-tail Student's t-test was performed with indicated p values

+HLA-DR+PD-1+ on CD8+ T cells can be associated with severe viral disease in the patients who died of influenza viral infection.

**HLA-A2-restricted M1₅₈+CD8+ T cells are CD38+HLA-DR+.** To verify that influenza-specific CD8+ T cells express the CD38+HLA-DR+PD-1+ phenotype, we analyzed CD8+ T cells specific for the immunodominant M1₅₈₋₆₆ epitope restricted by HLA-A*02:01 (A2-M1₅₈)[15]. In our H7N9 cohort, three of four patients expressing HLA-A*02:01 (a79, A9, and a10) had prominent IFNγ-CD8+ responses after H7N9 virus stimulation[5]. Longitudinal analysis of A2-M1₅₈+CD8+ T cells from a79 (Supplementary Fig. 1a, b) and a9 (Supplementary Fig. 1c, d), for total CD8+ T cells (Supplementary Fig. 1a, c) and the CD38+HLA-DR+CD8+ subset (Supplementary Fig. 1b, d) showed that the A2-M1₅₈+CD8+ population (in red) is predominantly CD38+HLA-DR+, CD38+PD-1+, and CD45RA−(Supplementary Fig. 1). The CD38+HLA-DR+PD-1+ CD8+ phenotype can indeed reflect antigen specificity and TCR ligation. However, as

only 13.6–34.9% (patient a79) and 8.98–16.8% (patient a9) of CD38+HLA-DR+CD8+ T cells were specific for the immunodominant A2-M1₅₈ peptide, the specificity status of the remaining CD38+HLA-DR+CD8+ T cells is unclear.

**Stable TCR repertoires within A2-M1₅₈+CD8+ T cells in H7N9.** To further define the molecular signatures of PBMC CD38+HLA-DR+CD8+ T cells from patients with different H7N9 disease outcomes, we dissected the clonal composition of both tetramer-specific A2-M1₅₈+CD8+ and total CD38+HLA-DR+CD8+ T-cell subsets. Here, utilizing direct ex vivo tetramer staining and human scTCRαβ multiplex RT-PCR[16,17], we show the first longitudinal dissection of TCRαβ repertoires for immunodominant A2+M1₅₈+CD8+ T cells[15,18] in PBMCs from three HLA-A*02:01-expressing patients (Fig. 3a; a9, a10, and a79) who survived H7N9 infection, as none of the patients with the fatal disease outcomes expressed HLA-A*02:01. Consistent with our earlier data for memory CD8+ T cells from healthy donors[15,19],

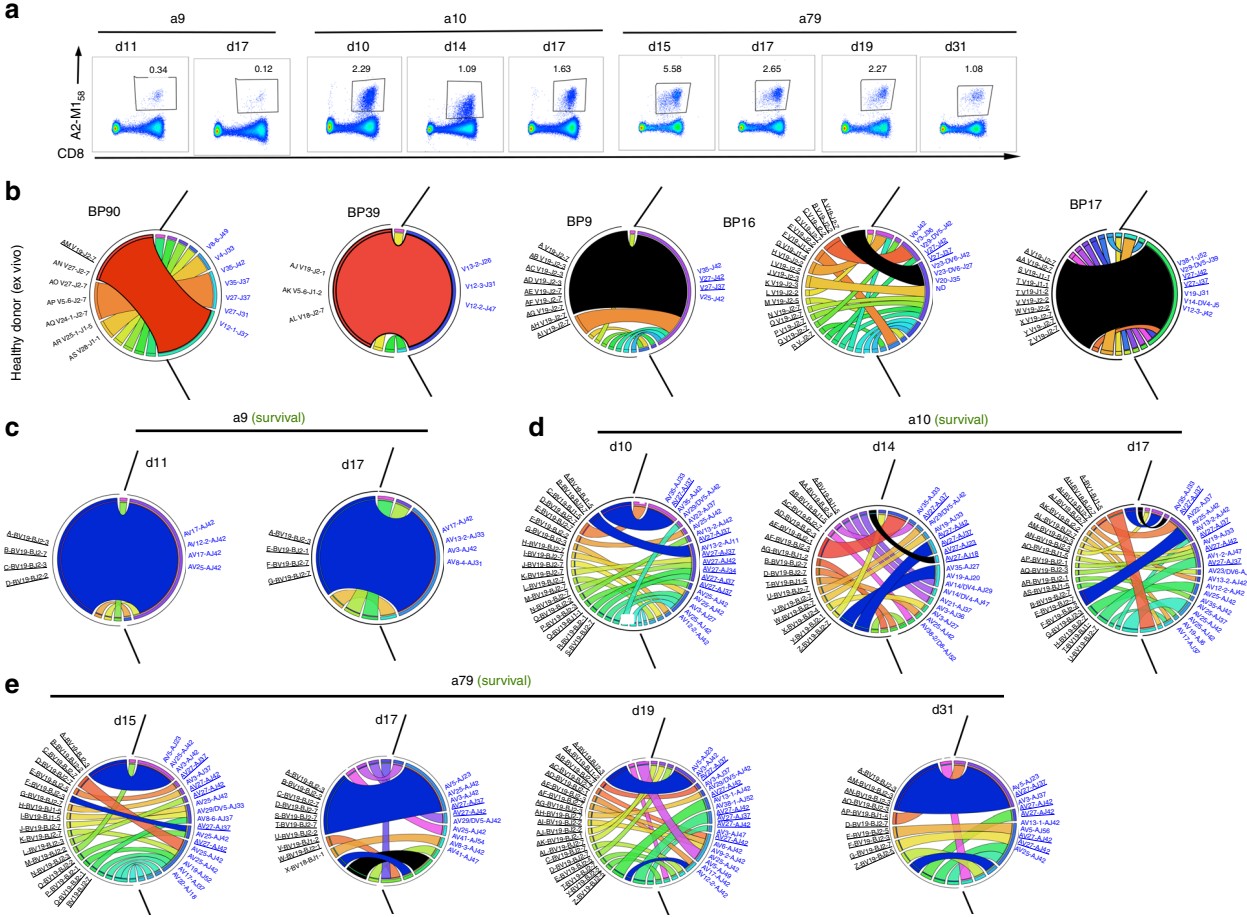

**Fig. 3** TCRαβ repertoire diversity and stability within A2-M1$_{58}$+CD8+ T cells in H7N9. **a** FACS profiles are shown for A2-M1$_{58}$+CD8+ T cells after tetramer staining directly ex vivo for a9, a10, and a79 patients. A2-M1$_{58}$+CD8+ T cells were single cell-sorted for TCRαβ repertoire analysis using a multiplex RT-PCR and sequencing. Populations were gated on viable Dump⁻tetramer+CD3+CD8+ events. Circos plots of frequencies of Vβ-Jβ (in black) and Vα-Jα (in blue) usage in paired TCRαβ sequences are shown for **b** memory A2-M1$_{58}$+CD8+ T cells in five healthy donors at one time-point or **c–e** activated A2-M1$_{58}$+CD8+ T cells in hospitalized patients (a9, a10, and a79) at different time-points after influenza infection. The width of the band is proportional to the frequency and each band represents a unique clone. Red segments represent private dominant A2-M1$_{58}$-specific TCR clones, black segments represent public A2-M1$_{58}$-specific TCR clones. Blue segments represent private dominant A2-M1$_{58}$-specific TCR clones present across different time-points in the H7N9 patients. Prominent TRBV19 and TRAV27 segments are underlined. Circos plots were generated with the Circos software package[40]

we found a bias towards TCRβ variable 19 (TRBV19) gene usage, coupled with a dominant TCRα variable 27 (TRAV27) (Fig. 3). The frequency of TRBV19 per patient repertoire across the time-points was 100% for patient a9 (Supplementary Table 1), 95% for a10 (Supplementary Table 2), and 99% for a79 (Supplementary Table 3). However, in contrast to memory A2-M1$_{58}$+CD8+ T cells from healthy individuals[15], in whom TRBV19 predominantly paired with TRAV27 (with a mean of 43.2%), the A2-M1$_{58}$+-specific TRBV19s detected at the acute phase of H7N9 infection commonly paired with a range of different TRAVs, with TRAV27 being generally found at lower frequencies such as 10% in a9, 20% in a10, and 23% in a79 T cells, (Fig. 3c–e). A unique feature of this H7N9 cohort was that a public common TCRαβ signature observed previously across healthy donors[15,20], with a predominant TCRαβ phenotype [complementarity-determining region (CDR3)β-SIRSSYEQ within TRBV19 and CDR3α-GGSQGNL within TRAV27], was found only in two H7N9 patients at very low frequencies, namely patient a10 at two time-points (at ~3–5%) and a79 at one time-point (13%), which is lower than the frequency in memory CD8+ T cells from healthy donors (15% for BP16, 62% for BP17, and 67% for BP9; Fig. 3b).

Rather, in all the surviving patients, large A2-M1$_{58}$+CD8+ T-cell clonal expansions were apparent early during infection and

persisted through subsequent time-points (Fig. 3c–e) bearing similar CDR3β and/or CDR3α motifs to the public TCR clonotype (Supplementary Table 1–3). In patient a9, the majority of the observed TCRαβ clones consisted of a single clonotype, "A", CDR3β-SMRSTDTQ/CDR3α-DGGGGSQGNL at d11 and d15 (>70%), with the common "RS" motif within the CDR3β and encompassing the public CDR3α-GGSQGNL sequence (Supplementary Table 1; Supplementary Fig. 2). In patient a10, two CDR3αβ clonotypes A, B were found at three time-points (d10, d14, and d17), while an additional seven CDR3αβ clonotypes were found at two time-points (Supplementary Table 2; Supplementary Fig. 2). Similarly, in patient a79, clonotype A CDR3β-SGRSADTQ/CDR3α-DNQGGKL was found at four time-points (d15, d17, d19, and d31) and an additional eight CDR3αβ clonotypes were observed at least at two time-points (Supplementary Table 3; Supplementary Fig. 2).

Thus, the tetramer-specific A2-M1$_{58}$+CD8+ TCRαβ repertoire is stable during acute H7N9 infection, with the clonal diversity that is at least comparable to that found for memory A2-M1$_{58}$+CD8+ T cells in healthy individuals (Fig. 3b)[15].

**Distinct clonal expansions within CD38+HLA-DR+CD8+ T cells.** Given differential CD38+HLA-DR+CD8+ T-cell

dynamics in H7N9 patients, we asked whether this was mirrored by distinct TCRαβ clonal composition and/or dynamics within CD38+HLA-DR+CD8+ T cells from five surviving (a9, a11, a20, a78, and a79) and two fatal (a33, a131) cases (Supplementary Table 4). As shown for a79 and a9 surviving patients expressing HLA-A*02:01 (Supplementary Fig. 3), the common TCRαβ clonotypes were shared between CD38+HLA-DR+CD8+ and A2-M1$_{58}$+CD8+ T cells within the same patient. While >50% of the A2-M1$_{58}$-specific TCRs were found with the CD38+HLA-DR+CD8+ population, only 5–18% of TCRs from all CD38+HLA-DR+CD8+ T cells were characteristic of the A2-M1$_{58}$+CD8+ repertoire (Supplementary Fig. 3), reflecting (as expected) greater clonotypic diversity within the "activated" CD38+HLA-DR+CD8+ set when compared to the single-specificity A2-M1$_{58}$+CD8+ T cells.

Further dissection of TCRαβ clonotypes in severe H7N9 influenza disease across different time-points showed a diverse usage of TCRα–TCRβ pairings within CD38+HLA-DR+CD8+ T cells, numerically comparable between the patients who survived (mean of 59.6 TRAV–TRBV different pairings, with a range 42–96; $n = 5$) and patients who died (mean of 68.5 TRAV–TRBV pairings; $n = 2$) (Fig. 4; Supplementary Table 5). As expected, a diversity of TRAV–TRBV pairings (mean of 62.1, $n = 7$ H7N9 patients) within the activated CD38+HLA-DR+CD8+ T cells (both HLA-A*02:01-expressing and HLA-A*02:01-non-expressing patients, Supplementary Table 6) was higher than that

within CD8+ T cells directed at a single A2-M1$_{58}$ epitope in H7N9-infected patients and healthy donors (mean of 7.9, $n = 8$; $p < 0.001$, standard two-tail Student's $t$-test) (Fig. 4d). However, interestingly, TCRαβ repertoires within CD38+HLA-DR+CD8+ T cells (Fig. 4; Supplementary Table 5) and A2-M1$_{58}$+CD8+ T cells (Supplementary Fig. 4; Supplementary Table 5) featured a similar number (~1–3) of clonally expanded TRAV–TRBV segments across H7N9 patients (Fig. 4; Supplementary Table 5), although larger TRAV–TRBV expansions were found within the single-specificity A2-M1$_{58}$+CD8+ T cells due to a limited number of TRAV–TRBV pairings (Supplementary Fig. 4).

Importantly, selected TCRαβ clonotypes were found more than once within CD38+HLA-DR+CD8+ T cells, within and across time-points (Supplementary Table 5), suggesting clonal expansions of specific TCRαβ CD38+HLA-DR+CD8+ clonotypes, thus their antigen specificity. Longitudinal analyses of TCRαβ clonal expansions showed that there was a significant effect of TCRαβ diversity over time across the patient group, as assessed by Simpson's Diversity Index (SDI) (Fig. 5a). In general, SDI values decreased in survivors with time after infection, but increased throughout the course of the disease for those who succumbed (DF = 9, $t = -2.3751$, $p = 0.0416$; $r^2$ of survival = 0.1405; $r^2$ of fatal = 0.3043, Simpson's Diversity Index (SDI))). Thus, while the survivors started with larger TCRαβ clonal expansions early on, this decreased with time as the patients recovered. Conversely, those who died started with lower (inefficient) TCRαβ clonal

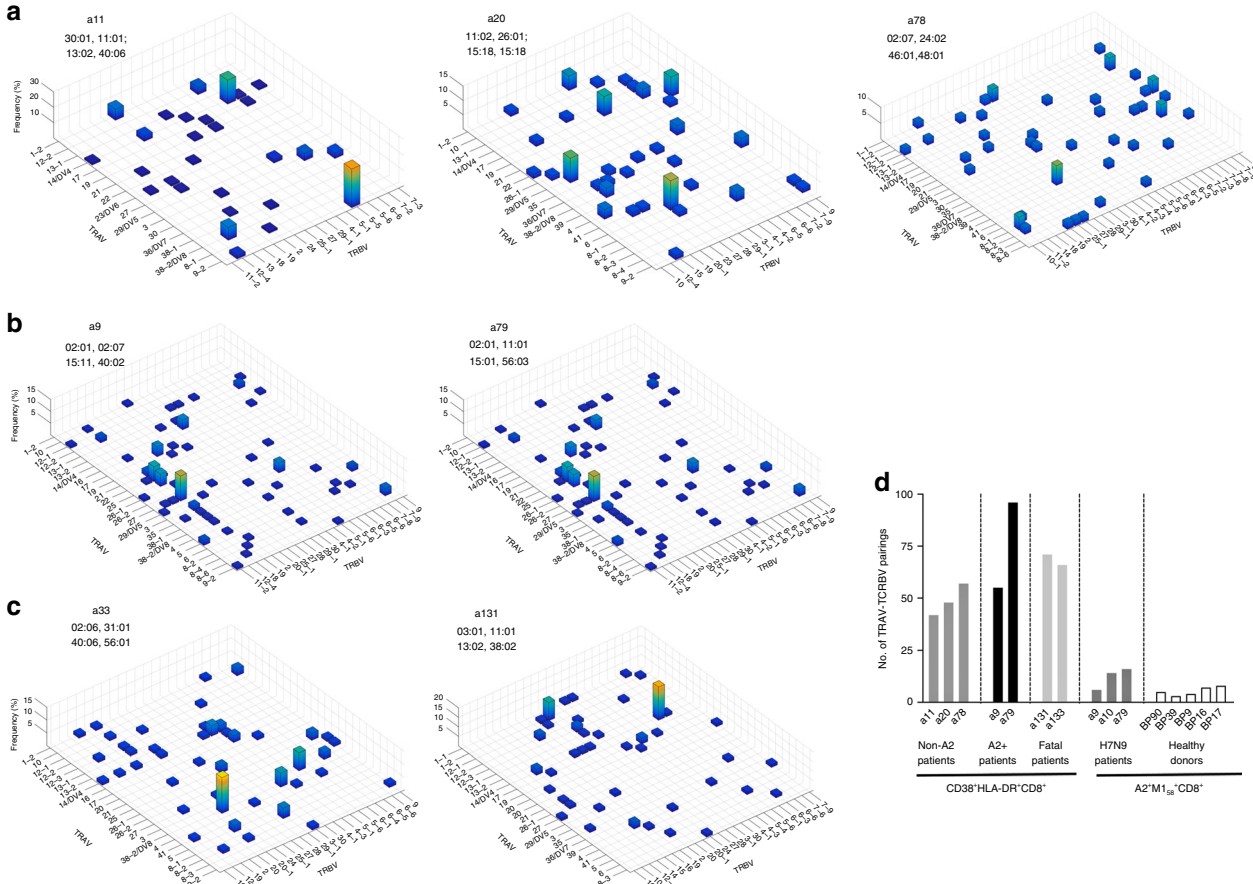

**Fig. 4** Diverse usage of TCRα–TCRβ pairings within CD38+HLA-DR+CD8+ T cells. The frequency of TCRαβ clonotypes within particular TRAV–TRBV paired segments is shown for individual patients: **a** non-HLA-A*02:01 surviving individuals; **b** HLA-A*02:01 surviving individuals; **c** patients with a fatal disease outcome. Pulled data from all the time-points is shown, as summarized in Supplementary Table 5. **d** The number of specific TRBV–TRAV paired segments utilized per donor is shown for non-HLA-A*02:01 surviving patients, HLA-A*02:01 surviving patients and fatal patients for CD38+HLA-DR+CD8+ T cells as well for H7N9 surviving patient and healthy individuals for A2-M1$_{58}$+CD8+ T cells

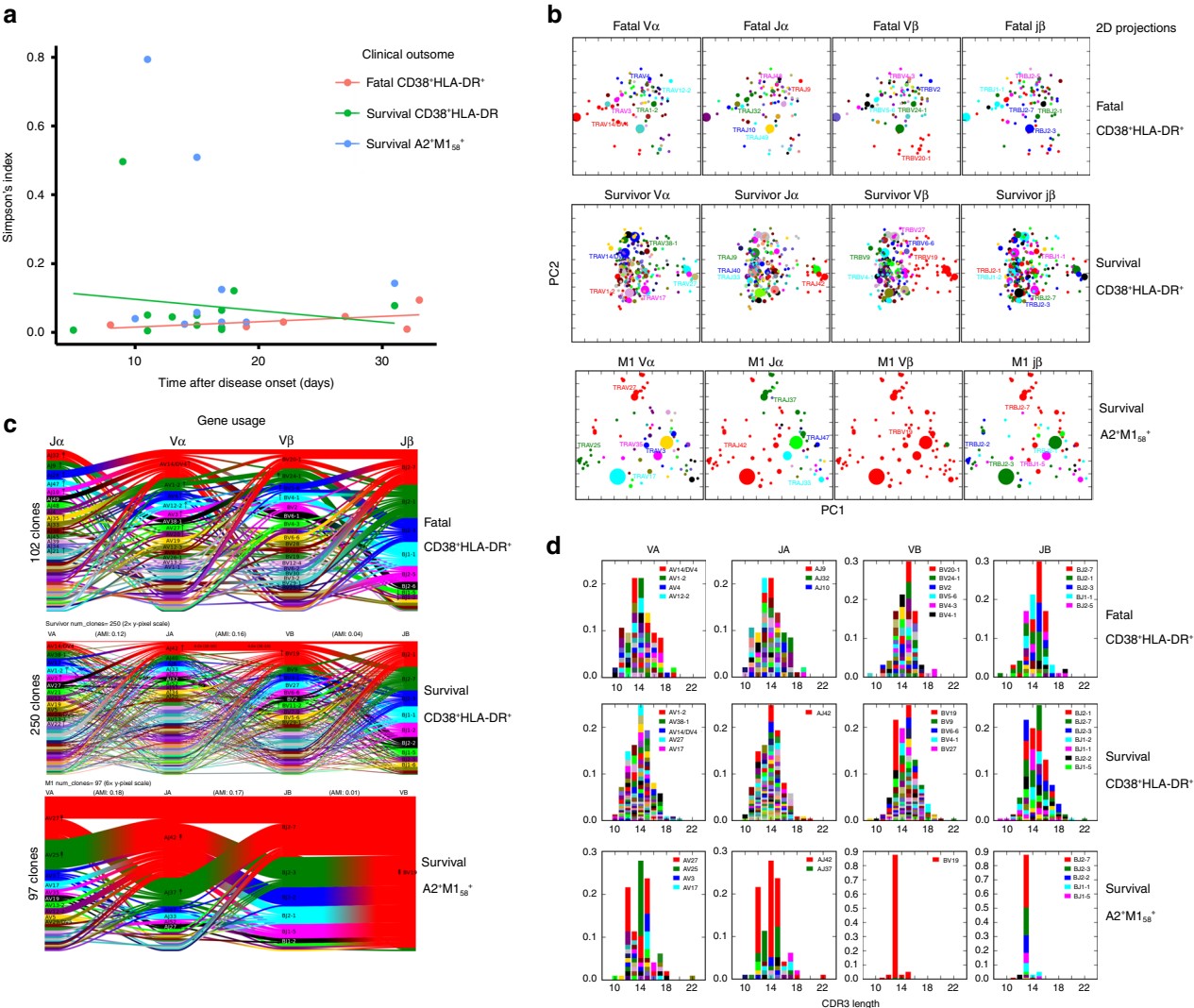

**Fig. 5** Distinct clonal expansion patterns within CD38$^+$HLA-DR$^+$CD8$^+$ T cells in H7N9. Longitudinal single-cell TCRαβ repertoire analysis of CD38$^+$HLA-DR$^+$CD8$^+$ T cells reveals delayed early clonal expansions within fatal patients. For comparison, TCRαβ characteristics of M1$_{58}$$^+$CD8$^+$ T cells (directed at a single influenza T cell specificity) are shown. **a** Simpson's Diversity Index (SDI) shows differential patterns of TCRαβ clonal expansions longitudinally within CD38$^+$HLA-DR$^+$CD8$^+$ T cells in patients who survived (in green) and died (in pink). SDIs for survival M1$_{58}$$^+$CD8$^+$ T cells are shown in blue. **b** Principal component analyses of Vα-chain segments, Jα regions, Vβ-chain segments, and Jβ segments (panels from left to right) were performed to identify increased clustering of TCRαβ elements for survival patients but more distinct TCR signatures for fatal cases, based on TCRdist analysis. The size of specific dot points reflects the size of specific TCR genes used. **c** V and J gene segment usage and pairing landscape is shown by average-linkage dendrogram of TCRdist clusters for Jα/Vα-chain segments and Jβ/Vβ-chain segments (panels from left to right). Segments are colored based on the frequency with which they occur within the repertoire with a color sequence, beginning with red (most frequent), then green (second most frequent), blue, cyan, magenta, and black. **d** CDR3 length against prominent Vα-chain segments, Jα regions, Vβ-chain segments, and Jβ regions is shown for CD38$^+$HLA-DR$^+$CD8$^+$ T cells within both the surviving and fatal H7N9 patients, as well as M1$_{58}$$^+$CD8$^+$ T cells within the surviving patients. The left right ordering of the segment types is chosen so that VA and JA are on the left, VB and JB are on the right, and the alpha–beta pairing with the largest adjusted mutual information is in the middle

expansions, which increased over time (Fig. 5a), concurrent with the greater prevalence of CD38$^+$HLA-DR$^+$CD8$^+$ T cells at later time-points (Fig. 1b). These results imply that a patient's recovery is associated with the early clonal expansion of a few "optimal" (most likely antigen-specific) CD38$^+$HLA-DR$^+$CD8$^+$ T cell clones, while the fatal cases are ultimately characterized by a delay in clonally expanded TCRαβ clonotypes within CD38$^+$HLA-DR$^+$CD8$^+$ T cells. We found no significant effects of the overall TCRαβ diversity on clinical outcome, with SDIs being: DF = 5, $t = 0.6071$, $p = 0.5703$ for the survival versus fatal groups.

Despite a broader range of TRAV–TRBV pairings used by CD38$^+$HLA-DR$^+$CD8$^+$ T cells, as compared to single-specificity

A2-M1$_{58}$$^+$CD8$^+$ T cells (Fig. 4d), the SDIs within CD38$^+$HLA-DR$^+$CD8$^+$ T cells and A2-M1$_{58}$$^+$CD8$^+$ T cells after H7N9 infection were similar (Fig. 5a). Three outliers were found within the SDI values, two within A2$^+$M1$_{58}$$^+$CD8$^+$ T cells and one outlier data point within CD38$^+$HLA-DR$^+$ CD8$^+$ T cells in the surviving patient a11 (Fig. 5a). In this patient, we found a largely expanded single clone (TRAV9-2-SQTGANNL and TRBV5-5-SSNHRAYYGY) constituting 71% of the CD38$^+$HLA-DR$^+$ TCRαβ repertoire at d9 (Supplementary Table 5). This clone decreased in size to 11% on d13 and was <4.5% (undetected) on d18. As this clonotype represents a repeated prominent TCRαβ found 11 times and 4 times at 2 different time-points (d9 and d13, Supplementary Table 5), this key TCRαβ signature was not

excluded from our analyses. TCRαβ analyses for additional patients and/or time-points could not be performed due to a limited number of longitudinal PBMC samples from this unique H7N9 cohort.

Our paired single-cell TCRαβ sequencing allowed us to dissect the further correlations between V and J segment usage within single TCRα or TCRβ chains (Vα-Jα, Vβ-Jβ) and across TCRαβ chains (Vα-Vβ, Vα-Jβ, and Vβ-Jα). Principal component analyses of TRAV, TRBV, Jα, and Jβ elements were performed to identify specific clustering segments specific to $CD38^+HLA-DR^+CD8^+$ T cells. Our data showed that TCRαβ clonotypes within $CD38^+HLA-DR^+CD8^+$ T cells displayed closely clustered signatures in the "recovery" group, but had much more diverse and dispersed features within the fatal group, again indicating that the early, substantial expansion of closely related TCRαβ clonotypes is indicative of an effective epitope-specific response (Fig. 5b). Again, in contrast to $CD38^+HLA-DR^+CD8^+$ T cells, $A2^+M1_{58}^+CD8^+$ T cells utilized fewer TCR elements, although these were associated with larger clonal expansions, as reflected by the size of the dot points within the selected motifs. This was in contrast to $CD38^+HLA-DR^+CD8^+$ T cells utilizing a wide range of TRAV, TRBV, Jα, and Jβ motifs. Most enriched in the survivor group was the "RS" motif from the CDR3β associated with the A2-M1_{58} epitope (Supplementary Fig. 5), although there were several other clusters defined by additional regions of unknown specificity.

Further dissection of the gene-segment pairing landscapes within $CD38^+HLA-DR^+CD8^+$ T cells and $A2^+M1_{58}^+CD8^+$ T cells were performed using TCRdist in a way that each clone was devoted a constant vertical height, while the curved segments joining neighboring gene stacks showed how the two gene distributions pair up, with the thickness of the segments corresponding to the number of clones having those two segments[21]. Significant gene–gene pairings (positive or negative correlations with a $P$-value less than 1e−6) were labeled at the beginning and ending of the corresponding segments. Gene–gene pairings, which were not observed and for which this under-representation was significant, are indicated by dashed segments with $P$-value labels. Enrichments (depletions) of gene segments relative to the background are shown for all labeled genes by up (down) arrows where the number of arrowheads reflects the base-2 logarithm of the fold change. While there were no striking differences between TCRαβ landscapes for $CD38^+HLA-DR^+CD8^+$ T cells across H7N9 patients and across all different time-points, the specific usage of particular TRAV, TRBV genes as well as Jα and Jβ segments for $M1_{58}^+CD8^+$ TCRαβ repertoire was strikingly different (Fig. 5c). These analyses also clearly demonstrated that, while the overall TCRαβ diversity was similar between $CD38^+HLA-DR^+CD8^+$ T cells and $A2^+M1_{58}^+CD8^+$ T cells (Fig. 5a), $A2^+M1_{58}^+CD8^+$ TCRαβ diversity is owed almost exclusively to chains other than Vβ (as depicted by the 4-headed arrow within BV19, which indicates that BV19 is log2(4-fold) enriched relative to background), whereas in $CD38^+HLA-DR^+CD8^+$ T cells we observed TCRαβ diversity that also included Vβ chains (with no large enrichments in the Vβ chains). Similarly, while there was a range of CDR3α and CDR3β lengths associated with $CD38^+HLA-DR^+$ T cells, these were much more skewed within $CD8^+$ T cells directed at a single-epitope A2-M1_{58} (Fig. 5d).

Overall, our dissections of activated $CD38^+HLA-DR^+$ TCRαβ repertoires during acute H7N9 infection highlight a broad usage of TCRαβ segments (TRBV, TRAV, Jβ, and Jα elements) in $CD38^+HLA-DR^+$ TCRαβ repertoires, with TCRαβ clonal expansions occurring earlier and were closely-clustered in the survival group, while delayed expansions with less TCRαβ gene clustering emerged in the fatal group.

**Single-cell RNAseq signatures for $CD38^+HLA-DR^+CD8^+$ T cells.** As our single-cell TCRαβ data within $CD38^+HLA-DR^+CD8^+$ T cells suggested differential expansion kinetics but similar TCRαβ clonotype diversity across TRAV–TRBV pairings and CDR3 lengths, we subsequently performed single-cell RNAseq (scRNAseq) to further dissect any underlying divergences at the gene transcript level. scRNAseq was performed for the total H7N9-driven $CD38^+HLA-DR^+CD8^+$ T cells from the surviving a11 (d15, d29) and fatal a33 (d12, d31) cases. Unsupervised principal component analysis of the overall $CD38^+HLA-DR^+CD8^+$ population established a clear segregation across the two patients, particularly at the early (a11–d15 and a33–d12) versus late (a11–d29 and a33–d31) time-points (Fig. 6a). In fact, $CD38^+HLA-DR^+CD8^+$ T cells from the surviving a11 individual at the earlier d15 time-point closely reflected $CD38^+HLA-DR^+CD8^+$ T cells from both the survival and fatal cases (a11–d29 and a33–d31) at the later time-points. By contrast, the gene profiles of the early a33–d12 "fatal" $CD38^+HLA-DR^+CD8^+$ T cells were distinct from any other population tested, suggesting a divergent differentiation pathway of this $CD38^+HLA-DR^+CD8^+$ set from the outset in this fatal patient. Interestingly, additional differentiation between expanded (found >1 time) versus non-expanded (found only 1 time) TCRαβ clones (Supplementary Table 7) further verified that there were global differences in transcriptomes and gene expression profiles with respect to patients' outcomes rather than expanded versus non-expanded TCRαβ clonotypes. In general, transcriptome profiles of expanded versus non-expanded TCRαβ clonotypes looked similar within a particular patient (Fig. 6a). This suggests either an early transcriptional divergence or an early recruitment of non-specific cells within $CD38^+HLA-DR^+CD8^+$ T cells in a patient with the fatal H7N9 disease outcome.

Similarly, using the scRNAseq data to assess TCRαβ clonotypes for their similarity within each patient across different time-points showed that, while cell-to-cell similarity was maintained over time for the surviving patient a11, there was a marked loss of similarity (or increase in transcriptomic heterogeneity) in the fatal patient a33 at the late stage of infection (Fig. 6b). These data are in agreement with the differential clonal expansions over time between patients who survived and died (Fig. 5a). Unsupervised hierarchical clustering and heatmap analysis identified 279 differentially expressed genes as four clusters segregating cells according to the patient and the time-point, with the effect being particularly evident for the Heat Shock Protein (DNAJB1), IFITM3, LDHA, RPA3, PSMA5, and PDCD5 genes in a fatal patient (Fig. 6c). Notably, genes that were identified showed differences between early and late time-points in the subject that died.

Although limited in numbers, we also analyzed gene profiles for all expanded and non-expanded TCRαβ clonotypes and found only minimal differences such as an increase in CCR5 and RNF8 expression within the expanded $CD38^+HLA-DR^+CD8^+$ clonotypes (Supplementary Fig. 6b). This further attests to the global differences in $CD38^+HLA-DR^+CD8^+$ transcriptomes between the surviving and fatal patients rather than the expansion profiles of those cells. It is important to note that our scRNAseq analysis only consists of two H7N9 patients over two time-points, due to the availability of those longitudinal samples.

Overall, our data show that there was an early developmental dichotomy in $CD38^+HLA-DR^+CD8^+$ transcriptomes between the surviving and fatal patients, associated with differential TCRαβ clonal expansion kinetics. This implies that "fatal" $CD38^+HLA-DR^+CD8^+$ T cells might be developmentally programmed to engage rapidly in anti-viral responses rather than antigen-specific TCRαβ-mediated $CD8^+$ T cell responses.

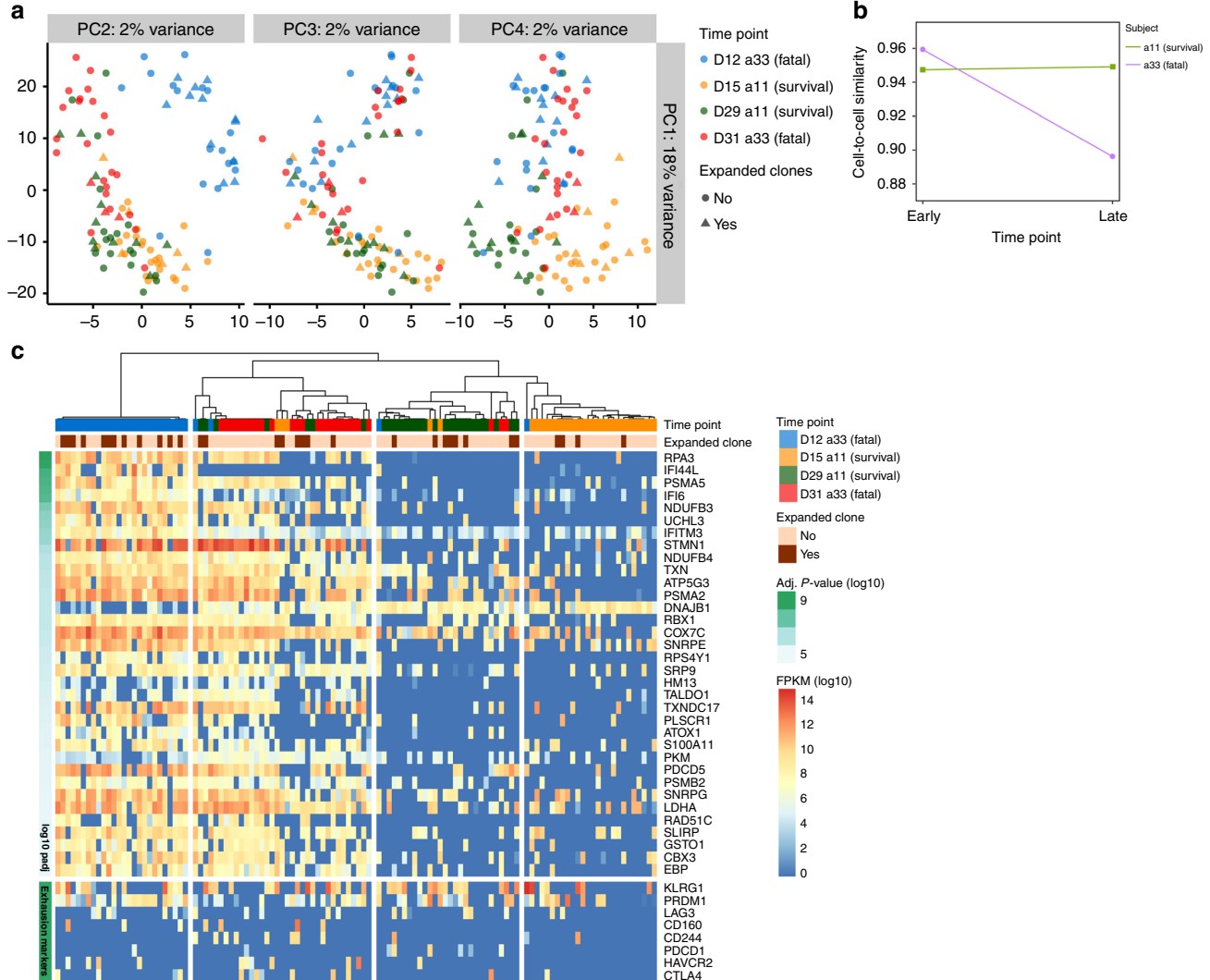

**Fig. 6** Global differences at the transcriptome levels within CD38+HLA-DR+CD8+ cells. Single-cell RNAseq (scRNAseq) was performed on CD38+HLA-DR+CD8+ T cells from longitudinal samples from two subjects, surviving a11 (d15, d29) and fatal a33 (d12, d31) patients. **a** Unsupervised principal-component analysis (PCA) of CD38+HLA-DR+CD8+ T cells reveals a clear segregation across the two patients, especially between survival (a11) and fatal (a33) at the early time-points (a11, d15 and a33, d12). Additional information between expanded versus non-expanded TCRαβ clonotypes (further defined in Supplementary Table 7) is shown using different symbols: triangles for expanded TCRαβ clonotypes and circles for non-expanded TCRαβ clonotypes; **b** RNAseq data were used to assess the similarity of CD38+HLA-DR+CD8+ T cells within each patient across different time-points. The average Pearson's correlation coefficient was calculated for each pair of CD38+HLA-DR+CD8+ T cells within the same time-point using FPKM values for all detected genes, revealing cell-to-cell similarity over time for surviving patient a11, and a marked loss of similarity (or increase in transcriptomic heterogeneity) in fatal patient a33 in the late stage of infection. **c** Unsupervised hierarchical single-cell consensus clustering (SC3) heatmap plot highlights the expanded TCRαβ clonotypes, patient, and time-points with fixed colors and legend. Differentially expressed genes and a priority list of genes associated with exhaustion show four clusters segregating cells according to the patient and the time-point (survival a11, d15, and d29; fatal a33, d12, and d31). FPKM: fragments per kilobase of transcript per million mapped reads

## Discussion

The fact that IAV infections are generally more severe in the elderly and in the young children variously reflects the general decline in host response efficacy with increasing age, versus the absence of any cross-reactive immune memory in those who have yet to be exposed to any IAV strain. With much of the response directed at IAV peptides from relatively conserved internal proteins, it is well established from experiments in both mice and ferrets[22–24] that cross-reactive CD8+ T-cell memory generated following exposure to serologically distinct IAVs can provide a measure of protection following challenge with a novel IAV. In addition, correlative data generated earlier for these hospitalized patients[5], showing varied disease outcomes (on a spectrum from rapid recovery to death) following infection with the avian H7N9 IAV, reinforces the case for protection mediated by previously primed IAV-specific CD8+ T-cell memory.

Building on these observations that immune CD8+ T cells mediate more rapid recovery from severe influenza[1,4,5,25], we further investigated the mechanisms underlying impaired functionality in patients with fatal disease outcomes versus survival following H7N9 infection[5]. While those who recovered had high and transient CD38+HLA-DR+PD-1+ expression, the fatal cases were characterized by prolonged CD38+HLA-DR+PD-1+ expression on CD8+ T cells that showed evidence of minimal IFNγ production. Many of these CD38+HLA-DR+PD-1+ T cells might not be IAV-specific, with this PBMC profile being

associated with ineffective, early expansion of CD8[+] T-cell clones expressing TCRαβs known to recognize IAV epitopes in the patients who died. Thus, lethal H7N9 virus, possibly via increased inflammation and prolonged viral exposure[2,3], causes striking pathophysiology with respect to CD8[+] T cells, as evident by high and prolonged CD38[+]HLA-DR[+]PD-1[+] expression, more reminiscent of persistent viral infections.

Utilizing the Shanghai H7N9-first wave longitudinal PBMC cohort, we previously found minimal evidence for the presence of IAV-specific, IFNγ-producing CD8[+] T cells in the fatal cases. Despite the absence of immune effectors in the patients who ultimately succumbed to H7N9 infection, a prominent characteristic of these "non-survival" PBMC populations was the high and prolonged frequency of an activated CD38[+]HLA-DR[+]CD8[+] set. Other studies of Ebola[8], HIV[9], dengue virus[10], and pandemic H1N1[26] infections have also mentioned the presence of such T cells, while the current, kinetic analysis clearly ties the prominence of this CD38[+]HLA-DR[+]CD8[+] phenotype to fatal outcomes in severe IAV pneumonia. In addition, this profile of CD38[+]HLA-DR[+] expression in lethal disease is further correlated with co-expression of the PD-1 exhaustion marker. Previously, high levels of PD-1 expression on CD8[+] T cells in HIV-1 progressors were associated with reduced IFNγ and perforin production, while long-term non-progressors displayed lower PD-1 expression and higher functionality[27]. The part played by PD-1 in CD8[+] T-cell exhaustion induced by continued TCR stimulation as a consequence of prolonged exposure to virus or tumor epitopes is well known[12–14,28,29], though this has not previously been understood for severe, acute infections. Indeed, it seems that two distinct patterns of PD-1 upregulation underlie different T-cell survival and functional outcomes depending on the nature of viral infection. While PD-1[int] CD8[+] T cells elicited by acute LCMV-Armstrong viral infection remained functional (IFNγ) and downregulated PD-1 (to PD-1[int]) after the virus was cleared, PD-1[hi]-expressing CD8[+] T cells displayed an exhausted phenotype in the face of chronic LCMV-clone-13 infection[14]. This PD-1[hi] phenotype was also characteristic of patients who died from severe H7N9 disease, suggesting that excess virus load in a normally acute disease process can mimic the profile characteristic of persistent infections like LCMV and HIV.

It remains unclear whether those CD38[+]HLA-DR[+]CD8[+] T cells that did not show evidence of binding (by IFNγ) to the IAV epitopes tested are antigen-specific (TCR-primed) or whether such CD8[+] T cells can express CD38[+]HLA-DR[+] as a consequence of some non-cognate, non-TCR-mediated stimulus. As ~50% of blood CD8[+] T cells from patients who succumbed to H7N9 were also CD38[+]HLA-DR[+], it seems unlikely that much of this activation profile was a consequence of direct TCR ligation. Conversely, Ndhlovu et al. have proposed that most CD38[+]HLA-DR[+] CD8[+] T cells found in acute HIV-1 infection are indeed HIV specific[9], as >40% of CD38[+]HLA-DR[+]CD8[+] T cells bound known HIV tetramers, though the antigen specificity of the remaining tetramer-negative CD8[+] set(s) was not determined.

Overall, relatively little is known about the longitudinal dynamics of TCRαβ selection and maintenance during human virus infections, especially acute infections. Recently, we showed the clonal stability during chronic CMV[30] and EBV[17] infections in the lung transplant patients. By contrast, detectable clonotypic turnover was found in HIV-specific CD8[+] T-cell populations during antiviral treatment or viral escape[31]. Our single-cell multiplex analysis of TCRαβ clonotypes within A2-M1$_{58}$ tetramer[+]CD8[+] T cells revealed clonal stability in survival cases during the course of acute H7N9 disease, as shown for influenza-specific CD8[+] T cells in mice[32,33]. The clonal diversity during acute H7N9 disease was comparable to the TCRαβ repertoire

characteristic of memory A2-M1$_{58}$[+]CD8[+] T cells in healthy individuals[15].

Further, analysis of "activated" CD38[+]HLA-DR[+]CD8[+] T cells within surviving and fatal patients established that the clonal expansion and dynamics reflected the profiles of CD38[+] HLA-DR[+]CD8[+] expression. While the patients who recovered from H7N9 had early, persistent, stable IAV-specific clonal expansions reflecting, at least in part, the reactivation of cross-reactive CD8[+] memory T cells, those with fatal outcomes lacked large clones early in the disease and their TCRαβ repertoires were constantly remodeling during the course of infection. Such differences were supported by global variations in gene profiles as shown by a divergent differentiation pathway of the CD38[+]HLA-DR[+]CD8[+] T cells from the outset in the fatal patient, suggesting that there was an early dichotomy in programming of CD38[+]HLA-DR[+]CD8[+] T cells in surviving and fatal patients.

Overall, the present analysis highlights the importance of establishing broadly cross-reactive memory TCRαβ clones that expand rapidly in numbers to control the infection and limit the extent of inflammatory damage. Conversely, though the early prominence of an activated CD38[+]HLA-DR[+]PD-1[+]CD8[+] set is associated with survival, the combination of lymphopenia with a high frequency of (perhaps) IAV-non-specific CD38[+]HLA-DR[+] PD-1[+]CD8[+] T cells is characteristic of fatal disease. Analyzing CD8[+] PBMC populations for prevalence of the CD38[+]HLA-DR[+] PD-1[+] phenotype along with determining the profiles of IAV-peptide induced IFNγ production might be informative in understanding the outcome of severe influenza disease. Future studies utilizing large patient cohorts might provide robust data whether such dysregulated and prolonged expression of CD38[+] HLA-DR[+] and PD-1[+] on peripheral blood CD8[+] T cells could predict disease severity and outcome.

## Methods

**Patient cohorts.** Patients infected with A/H7N9 influenza virus were hospitalized at the Shanghai Public Health Clinical Center (SHAPHC) and the clinical details have been published[5,7]. The disease course is described as the number of days after disease onset. Available longitudinal PBMCs from 11 H7N9-infected patients: eight patients who recovered (a9, a10, a11, a12, a130, a20, a78, and a79) and three patients with a fatal disease outcome (a22, a33, and a131) were analyzed. Informed consent was obtained from the participants and supervised by SHAPHC Ethics Committee. The study was approved by the SHAPHC Ethics Committee. Long-itudinal samples were also obtained from 22 patients hospitalized with PCR-confirmed seasonal influenza or 5 control patients hospitalized with non-viral infections at the Alfred Hospital during 2014–2016 (Dissection of Influenza Specific Immunity (DISI) cohort). Peripheral blood was taken at hospitalization, 3–5 days after hospitalization, at hospital discharge and ~30 days after symptom onset. Peripheral blood from the healthy individuals was collected on d0 before vaccination and on d7, d14, and d28 following vaccination with the inactivated influenza vaccination. The cohort included 18 individuals vaccinated with the 2015 trivalent (TIV) vaccine and 27 individuals vaccinated with the 2016 quadrivalent (QIV) vaccine. Informed consent was obtained from the participants and the study was approved by the Alfred Hospital Ethics Committee (280/14) and UOM Human Research Ethics Committee (1443389.3).

**Flow cytometric analysis.** PBMCs from H7N9 patients were stained with A2-M1$_{58}$ tetramer conjugated to PE (1:200) in FACS buffer (PBS with 1% bovine serum albumin). Cells were stained with antibodies: anti-CD3-PB (BD Cat #558117, UCTH1, 1:200), anti-CD8-FITC (BD Cat #347313, SK1, 1:50), anti-CD27-PE-Cy7 (eBioscience Cat #25-0279-42, O323, 1:25), anti-CD45RA-APC-Cy7 (BD Cat #560674, HI100, 1:50), anti-HLA-DR-ECD (Beckman Coulter Cat#IM3636, Immu-357, 1:25), anti-CD38-PerCPCy5.5 (Biolegend, Cat#303522, HIT2, 1:50), anti-PD-1-APC (Biolegend #329908, EH12.2H7, 1:25), and Live/Dead-aqua 525 (Invitrogen, 1:800). Lymphocytes were washed, acquired on a FACSAria II sorter with FACS Diva software (Becton Dickinson) and analyzed with FlowJo software (Treestar).

**Single-cell multiplex RT-PCR for CDR3β and CDR3α analysis.** CD8[+] T cells from patient samples were stained with an A2-M1$_{58}$ tetramer and A2-M1$_{58}$-tet-ramer[+]CD3[+]CD8[+] T cells were single-cell sorted (FACSAria II, BD Biosciences)

into 96-well plates. In selected experiments, CD38$^+$HLA-DR$^+$ CD3$^+$CD8$^+$ T cells were also single cell-sorted. The CDR3αβ regions were determined by a single-cell multiplex reverse transcription PCR (RT-PCR) protocol[15,16]. Positive PCR products were sequenced using TRAC or TRBC reverse internal primers (Supplementary Table 8) at the Sequencing and Genotyping Facility (Department of Pathology; University of Melbourne). Sequences were analyzed using FinchTV. V-J regions were identified by IMGT query (www.imgt.org/IMGT_vquest).

**Single-cell RNAseq.** Single-cell RNA sequencing libraries were prepared using the Smart-seq2 protocol as described[34,35], with the following modifications. Cells were sorted into 96-well plates containing 0.5 µl dNTP mix (10 mM), 0.5 µl oligo-dT primer (5 µM), and 1 µl lysis buffer (prepared by adding 1 µl RNase inhibitor to 19 µl Triton X-100 solution, 0.2% v/v). cDNA synthesis and pre-amplification reaction volumes were halved to 5 and 12.5 µl, respectively, and the ERCC RNA Spike-In Mix was added at a final concentration of 1:400 million to the reverse-transcription step.

**Data analysis of scRNA-seq.** Reads had an average length of 133 base pairs and each cell had an average sequencing-depth of 800,000 paired-end reads enough to robustly assess gene expression and reconstruct TCR sequences from scRNA-seq[36]. Quality control of scRNA-seq reads was performed with FastQC where reads were aligned to Ensembl GRCh37 reference genome using TopHat2[37] with default parameters. Gene expression was calculated in fragments per kilobases per million (FPKM) using Cufflinks suit (v2.2.1)[37]; more specifically CuffQuant was used to calculate FPKM and CuffNorm to normalize the FPKM values based on total mRNA content. Unsupervised hierarchical clustering was performed with the package SC3[38]. Heatmaps were generated with the R package made4 and pheatmap. SC3 package in R was utilized for clustering analysis, and to identify differentially expressed genes across clusters. This package was also utlized to visualize clusters and to link gene expression with TCR clonality. Gene ontology enrichment was performed with DAVID using differentially expressed genes with an adjusted $p$-value < 0.05 ($n = 277$). As background, the whole list of genes identified in the scRNA-seq data was used. Network analysis was performed to identify the predicted protein–protein interactions from genes identified via clustering (SC3). Basically, genes identified as classifying specific clusters in the SC3 analysis were utilized to identify the network of proteins associated to any known interactions. Networks of protein–protein association were performed with STRING DB[21].

**TCR analysis.** TCR sequences generated by Sanger sequencing were either called manually or using the Biopython algorithm and then parsed using the TCRdist analytical pipeline[39]. Clonotypes were defined as single-cell TCRαβ pairs that exhibited the same V, J, and CDR3 regions. In cases where TCRαβ pairs returned valid blast hits for all V and J regions but failed to return a valid CDR3 region due to sequencing errors, we manually called CDR3 regions using existing clonotypes from the same sample as a reference. SDI was calculated for each sample (i.e. patient-day combination) and analyzed as a function of clinical outcome (i.e. survival versus fatal), day (i.e. time since infection), and the interaction of these two variables. Specifically, we used a penalized quasi-likelihood approach to generalized linear mixed models (the glmmPQL function in the R MASS package)[40], with patient identity included as a random effect in order to account for the non-independence of samples collected from the same patient at different time-points. As SDI best fit a log-normal distribution, we modeled the data using a Gaussian distribution and a log link. The coefficient of determination ($r^2$) was independently calculated using the linear model (lm) function in R for each significant regression.

**Analysis of full length TCR from scRNAseq.** VDJPuzzle was utilized to reconstruct full-length TCRαβ from scRNAseq data[35]. To identify clonally expanded TCRs, we have considered a clone, a TCR when it was present in more than one cell across the available data in any samples time-point utilized for scRNAseq.

**Statistical analyses.** The absolute numbers of CD38$^+$HLA-DR$^+$CD8$^+$ T cells and the IFNγ$^+$CD8$^+$ T cells were compared between the survival group and the fatal cases using a Mann–Whitney test. The average CD8$^+$ T-cell counts were calculated by the ratio of the total area under the curve over 15 days (using trapezoidal integration). Numerical integration and least-squares fittings were implemented using MATLAB (version R2014b) (MathWorks, Natick, MA). Unless otherwise indicated, statistical differences of $p < 0.05$ were obtained as performed using a standard two-tail Student's $t$-test.

**Data availability.** The data that support the findings of this study are available from the corresponding author upon request. scRNAseq data that support the findings of this study have been deposited in Arrayexpress with the accession code E-MTAB-6379.

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

## Acknowledgements

We thank Janine Roney, Leah Christie, Jill Garlick for patient recruitment (Alfred Hospital), and Nicola Bird for technical assistance. This work was supported by NSFC 81471556, 81470094, 81430030, 81672018, Program for Emergency Response to H7N9 Outbreak (2013QLG003), Shanghai Municipal Commission of Health and Family Planning, and the 973 National Key Basic Research Project (2014CB542502), Ministry of Science and Technology of P.R. China, the University of Melbourne IRRTF Grant, and an NHMRC Program Grant (1071916) to K.K., N.L.L.G., S.J.T., and P.C.D. Z.W. was an NHMRC China-Australia Exchange Fellow, E.B.C., and A.A.E. are NHMRC Peter Doherty Fellows, F.L. is an NHMRC Career Development Fellow Level 2 Fellow, S.J.T. is an NHMRC Principal Research Fellow and K.K. is an NHMRC SRF Level B Fellow. S.Q. P. was a recipient of the University of Melbourne International Research Scholarship and a CONACyT Scholar. S.S. is supported by the Victoria-India Doctoral Scholarship (VIDS) and Melbourne International Fee Remission Scholarship (MIFRS). M.K. is a recipient of Melbourne International Research Scholarship and MIFRS. A.C. is supported by a NHMRC Career Development Fellowship.

## Author contributions

Designed experiments: Z.W., Y.W., S.M.Q.-P., L.Loh., K.K., and J.X. Performed experiments: Z.W., L.Z., T.H.O.N., Y.W., C.Q., S.M.Q.-P., A.A.E., M.K., E.B.C., L.Loh., L.Liu., Y.R., T.C., and X.Z. Analyzed experiments: Z.W., L.Z., T.H.O.N., S.S., Y.W., S.M.Q.-P., J. C.C., A.A.E., S.R., M.K., F.L., R.A.B., L.Loh., L.Liu., Y.R., P.C., L.K., P.G.T., P.C.D., K.K., and J.X. Establishment of clinical patient cohorts: Y.W., C.Q., Y.R., X.Z., T.C.K., A.C.C., and J.X. Wrote the manuscript: Z.W., Y.W., T.H.O.N., S.S., S.M.Q.-P., J.C.C., L.Liu., F.L., J.M.M., N.L.L.G., S.J.T., Y.R., P.G.T., P.C.D., K.K. and J.X.

## Additional information

**Competing interests:** The authors declare no competing financial interests.

