## [Peer Review File · Nature Communications]

Reviewer #1 (Remarks to the Author):

This manuscript describes the CD8 T cell responses in non-lethal vs lethal H7N9 influenza infection. The authors are in the unique position of having access to longitudinal blood samples on a rare cohort of H7N9 infected subjects, some of whom succumbed to their infection. The uniqueness of the cohort and the ability to study CD8 responses longitudinally in the cohort raises the interest in their findings to a high level. Unfortunately, the authors have not done a very good job of telling a compelling story, and one is left feeling that the conclusions fall far short of the potential for this type of study.

Specific criticisms:

1) The main conclusion from the manuscript (from the abstract and the introduction) is that monitoring for a dysfunctional (activated PD-1+) CD8 T cell phenotype during severe influenza infection is predictive of disease outcome. This seems like a strange conclusion, since there are certainly much better predictors (lymphopenia, respiratory parameters, etc). In addition, because of the small size of the cohort, it is unlikely that the presence of this phenotypic population can be used to predict outcome (ie the sensitivity, specificity, and positive predictive values are unknown). It would seem that the findings would be better interpreted in terms of what they suggest about the pathophysiology of lethal H7N9 infection with specific respect to the CD8 T cell response.

2) Figures 1e-i evaluate the CD8 T cell response in comparison to length of hospital stay or lethality. In fact, all of the comparisons should probably be done as a binary comparison – survival vs fatal. It is unclear what factors went into deciding on the length of hospital stay, and all of the survivors and all of the fatalities cluster together very closely in terms of hospital stay length.

3) As shown in Figures 1b-d, the CD8 T cell populations change with time of infection. It is unclear what values (from which time points) were used in Figures e-i. The description may already be in the manuscript somewhere, but this reviewer could not find the information, so it needs to be more clearly stated somewhere.

4) Much is made of the changes in TCR repertoire diversity in fatal vs non-fatal infection. These data are shown in Figures 6a-c. There are several problems. First, in Figure 6a the conclusion that the Simpson's index is decreasing in survivors appears to be driven by a single outlier data point (SDI of 0.5). Does the conclusion still hold without this one data point? Second, the data in Figure 6b are uninterpretable. The figure legend gives little information on what is being shown, the axes are not labelled, and the plots are so small that nothing can be read within each plot. Third, pastel colors are used in the PCA in Figure 6c and the symbols are so small that neither the colors nor the shapes can be discerned. Again, this figure is uninterpretable. Finally, most analyses are now being done with tsNE rather than PCA – has this been tried?

Overall, the manuscript is based upon a unique set of patients, but the message is unclear. In addition, it meanders from studying an activated CD8 population in patients, then jumps to a study of a mouse model, then jumps back to the humans for an assessment of TCR diversity. The authors need to re-think the message, what data are needed to solidify that message, and how best to present the necessary data. Anything that is not necessary should be eliminated.

Reviewer #2 (Remarks to the Author):

This manuscript reported the prolonged active CD38+/DR=/PD-1+ CD8 T cells in fatal cases of H7N9 infected individual. An excellent research team with combined expertise especially influenza virus infection and unique samples set. However, I have some major concerns as the following:

1. CD38+/DR+ phenotype in association with T cell activation is a well-known fact and PD-1+ cell is known to be expressed on highly active cells.

2. By sorting this group cells for RNAseq analysis as the result shown mainly confirmed those group of T cells are highly activated which evidenced by molecular signatures associated with cell

activation and T cells from fatal case are more activated as expected. It would be much more interesting if the TCR clonal type analysis could be included to further dissect the functional differences in expanded and not expanded T cells. However conclusions drawn by comparing one patient in each group is premature.

3. I also have concerns with the mouse experiment and how that would help the understanding of H7N9 human infection apart from the confirmation of T cell activation is associated with flu infection

4. The most interesting experiment in this paper is the T cell receptor repertoire analysis which is novel, i.e stable TCR repertoire of A2 M58-66 specific T cells during acute H7N9 infection, however the conclusion of ineffective early clonal expansion is premature and only drawn from limited number of patients (5 survival vs 2 fatal cases) with diverse HLA background are not solid (with two survivals with bearing HLA-A2 with a well know clonal expanded population which in my view should be excluded for analysis)

RESPONSES TO REVIEWERS' COMMENTS

We thank the Reviewers for their constructive comments and appreciating the importance of our study: “The uniqueness of the cohort and the ability to study CD8 responses longitudinally in the cohort raises the interest in their findings to a high level.” (Reviewer 1); “An excellent research team with combined expertise especially influenza virus infection and unique samples set.” (Reviewer 2).

We responded to Reviewers' questions and comments in a point-by-point form.

We appreciate the opportunity to re-submit our study to *Nature Communications* and are happy to make any additional modifications suggested by the Editors and Reviewers.

REVIEWER 1:

This manuscript describes the CD8 T cell responses in non-lethal vs lethal H7N9 influenza infection. The authors are in the unique position of having access to longitudinal blood samples on a rare cohort of H7N9 infected subjects, some of whom succumbed to their infection. The uniqueness of the cohort and the ability to study CD8 responses longitudinally in the cohort raises the interest in their findings to a high level. Unfortunately, the authors have not done a very good job of telling a compelling story, and one is left feeling that the conclusions fall far short of the potential for this type of study.

Following the Reviewer's comment, we have performed additional analyses of our TCR $\alpha\beta$ data and re-written the manuscript to tell a compelling story. As noted by both Reviewers, our TCR data are most novel and unique. Thus, we particularly analysed in depth and highlighted our TCR findings in the modified version of the manuscript.

Specific criticisms:

1) The main conclusion from the manuscript (from the abstract and the introduction) is that monitoring for a dysfunctional (activated PD-1+) CD8 T cell phenotype during severe influenza infection is predictive of disease outcome. This seems like a strange conclusion, since there are certainly much better predictors (lymphopenia, respiratory parameters, etc). In addition, because of the small size of the cohort, it is unlikely that the presence of this phenotypic population can be used to predict outcome (ie the sensitivity, specificity, and positive predictive values are unknown). It would seem that the findings would be better interpreted in terms of what they suggest about the pathophysiology of lethal H7N9 infection with specific respect to the CD8 T cell response.

Based on the Reviewer's comment, we have toned down the proposed role of CD38⁺HLA-DR⁺PD-1⁺ expression as predictive markers for disease outcomes (Abstract, Introduction page 3, Discussion page 10). We have suggested that “future studies utilizing large patient cohorts might provide robust data whether such dysregulated and prolonged expression of CD38⁺HLA-DR⁺ and PD-1⁺ on peripheral blood CD8⁺ T cells could predict disease severity and outcome” (Discussion, page 10).

We have also emphasized that lethal H7N9 virus, possibly via increased inflammation and prolonged viral exposure causes striking pathophysiology with respect to CD8⁺ T cells, as evident by high and prolonged CD38⁺HLA-DR⁺PD-1⁺ expression, more reminiscent of

persistent infection (Discussion, page 10).

“Lethal H7N9 virus, possibly via increased inflammation and prolonged viral exposure^{2,3} causes striking pathophysiology with respect to CD8⁺ T cells as evident by high and prolonged CD38⁺HLA-DR⁺PD-1⁺ expression, more reminiscent of persistent viral infection.”

2) Figures 1e-i evaluate the CD8 T cell response in comparison to length of hospital stay or lethality. In fact, all of the comparisons should probably be done as a binary comparison – survival vs fatal. It is unclear what factors went into deciding on the length of hospital stay, and all of the survivors and all of the fatalities cluster together very closely in terms of hospital stay length.

We have performed binary comparisons for survival vs fatal groups for **Fig.1h** and **Fig.1i** (shown also below), as per Reviewer’s suggestion. We used those figures in the modified version of our manuscript. Our conclusions and statistical significance remained unchanged with the binary analyses.

Additionally, we have performed univariate logistic regression on outcome (survival versus fatal) for the three independent variables #CD38⁺CD8⁺, #IFN γ ⁺CD8⁺ and the length of hospital stay. These analyses confirmed the reported findings that the disease outcome correlated to the #IFN γ ⁺CD8⁺ T cells (**Fig.1i**), but not #CD38⁺HLA-DR⁺CD8⁺ T cells (**Fig.1h**). As these alternative statistical analyses provide no additional information, they have not been noted in the manuscript.

Figure 1hi. Binary comparisons between the survival and fatal groups were performed for the hospital stay time (in days) versus (h) numbers of activated CD38⁺HLA-DR⁺CD8⁺ T-cells in 1 ml of blood, as peak values of all the assayed time-points, are shown for patients who survived versus patients who died; and (i) numbers of activated IFN γ ⁺CD8⁺ T-cells in 1 ml of blood, as peak values of all the assayed time-points, are shown for patients who survived versus patients who died.

3) As shown in Figures 1b-d, the CD8 T cell populations change with time of infection. It is unclear what values (from which time points) were used in Figures e-i. The description may already be in the manuscript somewhere, but this reviewer could not find the information, so

it needs to be more clearly stated somewhere.

We have clarified Figure 1 in the manuscript (Results, page 4) and in the figure legend:

Fig 1e: frequencies of activated CD38⁺HLA-DR⁺CD8⁺ T-cells, as mean values of all the assayed time-points, are shown for patients who survived versus patients who died;

Fig 1f: numbers of CD38⁺HLA-DR⁺CD8⁺ T-cells (data from all the assayed time-points) in 1ml of blood was analyzed according to the disease outcome;

Fig 1g: numbers of IFN γ ⁺CD8⁺ T-cells in 1ml of blood (data from all the assayed time-points) was analyzed according to the disease outcome;

Fig 1h: numbers of activated CD38⁺HLA-DR⁺CD8⁺ T-cells in 1 ml of blood, as peak values of all the assayed time-points, are shown for patients who survived versus patients who died;

Fig 1i: numbers of activated IFN γ ⁺CD8⁺ T-cells in 1 ml of blood, as peak values of all the assayed time-points, are shown for patients who survived versus patients who died;

4) Much is made of the changes in TCR repertoire diversity in fatal vs non-fatal infection. These data are shown in Figures 6a-c. There are several problems. First, in Figure 6a the conclusion that the Simpson's index is decreasing in survivors appears to be driven by a single outlier data point (SDI of 0.5). Does the conclusion still hold without this one data point?

We agree with the Reviewer regarding the presence of an outlier data point within CD38⁺HLA-DR⁺ CD8⁺ T cells in Figure 6a (**Fig.5a** in the modified version of the manuscript). This outlier comes from a largely expanded single clone (TRAV9-2-SQTGANNL and TRBV5-5-SSNHRAYYG) constituting 71% of the TCR $\alpha\beta$ repertoire at d9 in survival patient a11 (Supplementary Table S5). This clone decreased in size to 11% on d13 and was <4.5% (undetected) on d18.

Prior to the initial submission of this manuscript, we have had several discussions about this outlier, and possibility of its exclusion, but as this clonotype represents a repeated prominent clonotype found 11 times and 4 times at 2 different time-points (d9 and d13, Supplementary Table S5), we had no scientific basis to exclude this key TCR $\alpha\beta$ signature from our analyses.

However, following the Reviewer's comment, we have performed additional longitudinal analyses of TCR $\alpha\beta$ clonotype expansions and diversity within A2⁺M1₅₈⁺ CD8⁺ T cells from H7N9 patients and included these data in **Fig.5a**. Our analyses demonstrate that in comparison to clonal expansions within A2⁺M1₅₈⁺CD8⁺ T cells, the CD38⁺HLA-DR⁺ clonal 'outlier' is within a range of CD8⁺ T cell clonal distributions, rather than an outlier.

As we acknowledge in the manuscript, we had limited longitudinal PBMC samples obtained from this unique H7N9 patient cohort. Thus, removing the whole data-point (outlier), changes the relationship between survival and fatal patients. Unfortunately, TCR $\alpha\beta$ analyses for additional patients and/or time-points could not be performed due to a limited number of longitudinal PBMC samples from this unique H7N9 cohort.

In the modified version of our manuscript, we have included the above information on the TCR $\alpha\beta$ clonotype (outlier) in a11 patient, explained that we have no scientific basis to exclude this outlier and stated the limitations of our analyses due to a limited number of

longitudinal PBMC samples obtained from this unique H7N9 patient cohort (Results; page 7):

“Despite a broader range of TRAV-TRBV pairings used by CD38⁺HLA-DR⁺CD8⁺ T-cells, as compared to single-specificity A2-M1₅₈⁺CD8⁺ T cells (**Fig.4d**), the SDIs within CD38⁺HLA-DR⁺CD8⁺ T cells and A2-M1₅₈⁺CD8⁺ T cells after H7N9 infection were similar (**Fig.5a**). Three outliers were found within the SDI values, two within A2⁺M1₅₈⁺CD8⁺ T cells and one outlier data point within CD38⁺HLA-DR⁺CD8⁺ T cells in the surviving patient a11 (Fig.5a). In this patient, we found a largely expanded single clone (TRAV9-2-SQTGANNL and TRBV5-5-SSNHRAYYGY) constituting 71% of the CD38⁺HLA-DR⁺TCRαβ repertoire at d9 (Table S5). This clone decreased in size to 11% on d13 and was <4.5% (undetected) on d18. As this clonotype represents a repeated prominent TCRαβ found 11 times and 4 times at 2 different time-points (d9 and d13, **Table S5**), this key TCRαβ signature was not excluded from our analyses. TCRαβ analyses for additional patients and/or time-points could not be performed due to a limited number of longitudinal PBMC samples from this unique H7N9 cohort.”

Figure 5. Differential early clonal expansion patterns within CD38⁺HLA-DR⁺CD8⁺ T-cells in patients who survived or died from H7N9. Longitudinal single-cell TCRαβ repertoire analysis of CD38⁺HLA-DR⁺CD8⁺ T-cells reveals delayed early clonal expansions within fatal patients. For comparison, TCRαβ characteristics of M1₅₈⁺CD8⁺ T-cells (directed at a single influenza T cell-specificity) are shown. (a) Simpson’s Diversity Index (SDI) shows differential patterns of TCRαβ clonal expansions longitudinally within CD38⁺HLA-DR⁺CD8⁺ T-cells in patients who survived (in green) and died (in pink). SDIs for survival M1₅₈⁺CD8⁺ T-cells are shown in blue.

We have also toned down our conclusions (Title, Abstract, Results, Discussion) and re-analysed the data (new **Fig.4**, **Fig.5** and **Fig.6**; please refer to the manuscript) to provide a thorough analysis of our novel dissections of activated CD38⁺HLA-DR⁺ TCR αβ repertoires during acute viral infection, in comparison to TCRαβ repertoires obtained from CD8⁺ T cells directed at a single A2⁺M1₅₈ epitope (modified Results, pages 7-9). Our new comparisons show similar TCRαβ diversities but highlight significantly broader TCRαβ gene usage within CD38⁺HLA-DR⁺ TCR αβ repertoires (as compared to single-specificity M1₅₈⁺CD8⁺ TCRαβs), with closely-clustered TCRαβ segments (TRBV, TRAV, Jβ and Jα elements), detected in surviving but not fatal patients.

New title:

Prolonged persistence of clonally-diverse CD38⁺HLA-DR⁺CD8⁺ T cells during fatal H7N9 disease

Second, the data in Figure 6b are uninterpretable. The figure legend gives little information on what is being shown, the axes are not labelled, and the plots are so small that nothing can be read within each plot.

We have provided additional details in Results (page 8) and the figure legend to ensure **Fig.6b** (**Fig.5b** in the modified version of the manuscript) is easy to interpret:

“Our paired single-cell TCR $\alpha\beta$ sequencing allowed us to dissect further correlations between V and J segment usage within single TCR α or TCR β chains (V α -J α , V β -J β) and across TCR $\alpha\beta$ chains (V α -V β , V α -J β , V β -J α). Principal component analyses of TRAV, TRBV, J α and J β elements were performed to identify specific clustering segments specific to CD38⁺HLA-DR⁺CD8⁺ T cells. Our data showed that TCR $\alpha\beta$ clonotypes within CD38⁺HLA-DR⁺CD8⁺ T-cells displayed closely-clustered signatures in the ‘recovery’ group, but had much more diverse and dispersed features within the fatal group, again indicating that the early, substantial expansion of closely-related TCR $\alpha\beta$ clonotypes is indicative of an effective epitope-specific response (**Fig.5b**). Again, in contrast to CD38⁺HLA-DR⁺CD8⁺ T-cells, A2⁺M1₅₈⁺CD8⁺ T cells utilized fewer TCR elements, although these were associated with larger clonal expansions, as reflected by the size of the dot-points within the selected motifs. This was in contrast to CD38⁺HLA-DR⁺CD8⁺ T cells utilizing a wide range of TRAV, TRBV, J α and J β motifs. Most enriched in the survivor group was the “RS” motif from the CDR3 β associated with the A2-M1₅₈ epitope (**Fig.S5**), although there were several other clusters defined by additional regions of unknown specificity.”

We have enlarged **Fig.5b** and labelled the axes. We have also modified the graph so the size of the specific TCR dot-points depicts the size of specific TCR elements used.

We have also made modifications to **Fig.5b** to include TCR $\alpha\beta$ data on A2⁺M1₅₈⁺CD8⁺ T cells to highlight striking differences between TCRs within activated CD38⁺HLA-DR⁺CD8⁺ T cells during human influenza virus infection (our novel findings) and within single specificity A2⁺M1₅₈⁺CD8⁺ T cells.

Figure 5b. Principal component analyses of V α -chain segments, J α regions, V β -chain segments and J β segments (panels from left to right) were performed to identify increased clustering of TCR $\alpha\beta$ elements for survival patients but more distinct TCR signatures for fatal cases, based on TCRdist analysis. The size of specific dot-points reflects the size of specific TCR genes used.

Third, pastel colors are used in the PCA in Figure 6c and the symbols are so small that neither the colors nor the shaped can be discerned. Again, this figure is uninterpretable.

We have changed pastel colours to bright colours in **Fig.6c** (**Fig.6a** in the modified version of the manuscript) and additionally differentiated between expanded versus non-expanded clones using different symbols: triangles for expanded clones and circles for non-expanded clones.

Figure 6. Single-cell RNAseq profiles for CD38⁺HLA-DR⁺CD8⁺ T-cells reveal global differences at the transcriptome levels between a surviving and fatal patient. Single-cell RNAseq (scRNAseq) was performed on CD38⁺HLA-DR⁺CD8⁺ T-cells from longitudinal

samples from two subjects, surviving a11 (d15, d29) and fatal a33 (d12, d31) patients. (a) Unsupervised principal-component analysis (PCA) of CD38⁺HLA-DR⁺CD8⁺ T-cells reveals a clear segregation across the two patients, especially between survival (a11) and fatal (a33) at the early time-points (a11 d15 and a33 d12). Additional information between expanded versus non-expanded TCRαβ clonotypes (further defined in **Table S7**) is shown using different symbols: triangles for expanded TCRαβ clonotypes and circles for non-expanded TCRαβ clonotypes;

Finally, most analyses are now being done with tsNE rather than PCA – has this been tried?

Following the Reviewer's comment, we have performed additional dimensionality reduction of our scRNAseq via PCA (**Fig.6a** in the modified version of the manuscript) and tSNE (**Fig.s6**) approaches.

PCA analysis (as shown in the figure above) clearly revealed distinct subsets of CD38⁺HLA-DR⁺CD8⁺ T cells. There was a striking separation between CD38⁺HLA-DR⁺CD8⁺ T cells identified at d12 (early time-point) in the subject (a33) with a fatal disease outcome from the rest of the samples (please refer to the analysis using the first two dimensions of the PCA in **Fig.6a**). Further investigation of the third and fourth dimension of the PCA analysis also revealed distinct gene profiles between the two subjects a33 fatal patient at early and late time-points versus a11 surviving patient at early and late time-points (PCA1 with PCA3 and PCA4 in **Fig.6a**). Notably, there was no clear separation of T cells that were clonally expanded (depicted with triangles).

Finally, similar dimensionality reduction analysis with tSNE was performed, however as tSNE approach requires large datasets, while PCA can deal with smaller data sizes, we chose to include the PSA plots in the manuscript and tSNE analysis in supplementary Fig.S6.

Overall, the manuscript is based upon a unique set of patients, but the message is unclear. In addition, it meanders from studying an activated CD8 population in patients, then jumps to a study of a mouse model, then jumps back to the humans for an assessment of TCR diversity. The authors need to re-think the message, what data are needed to solidify that message, and how best to present the necessary data. Anything that is not necessary should be eliminated.

Following the Reviewer's comment, we eliminated mouse data from our manuscript and extended our TCRαβ analyses to solidify the conclusions and compare TCRαβ characteristics within CD38⁺HLA-DR⁺CD8⁺ T cells during acute influenza disease with those of single T cell specificity A2-M1₅₈⁺CD8⁺ T cells. Our manuscript provides the first data on TCRαβ repertoire dissection CD38⁺HLA-DR⁺CD8⁺ T cells. We have modified the manuscript and figures accordingly.

REVIEWER 2:

This manuscript reported the prolonged active CD38⁺/DR⁺/PD-1⁺ CD8 T cells in fatal cases of H7N9 infected individual. An excellent research team with combined expertise especially

influenza virus infection and unique samples set. However, I have some major concerns as the following:

1. CD38⁺/DR⁺ phenotype in association with T cell activation is a well-known fact and PD-1⁺ cell is known to expressed on highly active cells.

Although CD38⁺HLA-DR⁺ phenotype is known to be associated with T cell activation, to the best of our knowledge, we are the first to demonstrate prolonged and dysregulated kinetics of CD38⁺HLA-DR⁺PD-1⁺ on CD8⁺ T cells during severe viral disease in patients who died. Furthermore, we are the first to dissect TCRαβ repertoires within CD38⁺HLA-DR⁺PD-1⁺ and provide understanding on the extent of clonal diversity and expansions during influenza virus infection. Following the comments from both Reviewers', we focused our message on these novel data and importantly, performed new analyses to compare clonal TCRαβ diversity and molecular signatures within the activated CD38⁺HLA-DR⁺ CD8⁺ T cells with influenza-specific CD8⁺ T cells of one specificity, represented by HLA-A2⁺M1₅₈⁺ CD8⁺ T cells. Please see below new **Fig.5cd**:

Figure 5. (c) V and J gene segment usage and pairing landscape is shown by average-linkage dendrogram of TCRdist clusters for Vα-chain segments, Jα regions, Vβ-chain segments and Jβ segments (panels from left to right). Segments are coloured by frequency they occur within the repertoire with a colour sequence, beginning with red (most frequent), then green (second most frequent), blue, cyan, magenta, and black. (d) CDR3 length against prominent Vα-chain segments, Jα regions, Vβ-chain segments and Jβ regions is shown for CD38⁺HLA-DR⁺CD8⁺ T-cells within both surviving and fatal H7N9 patients as well as M1₅₈⁺CD8⁺ T-cells within surviving patients. The left-right ordering of the segment types is chosen so that Vα and Jα are on the left, Vβ and Jβ are on the right, and the alpha-beta pairing with the largest adjusted mutual information is in the middle.

2. By sorting this group cells for RNAseq analysis as the result shown mainly confirmed those group of T cells are highly activated which evidenced by molecular signatures associated with cell activation and T cells from fatal case are more activated as expected. It would be much more interesting if the TCR clonal type analysis could be included to further dissect the functional differences in expanded and not expanded T cells. However,

conclusions drawn by comparing one patient in each group is premature.

Following the Reviewer's comment, we divided CD38⁺HLA-DR⁺CD8⁺ TCRs into (i) clonally expanded TCRs (defined as TCR clones found >1 time and (ii) non-expanded TCRs (found only 1 time), as described in new **Table S7**). We have depicted those expanded clones on the PCA plots as triangles (**Fig.6a**) and on the scRNAseq heat-map (**Fig.6c; Supplementary Fig 6b**). As shown by our data (**Fig. 6abc**, please refer to the manuscript), there were global differences in transcriptomes and gene expression profiles between surviving versus fatal patients rather than between expanded versus non-expanded TCR clones. In general, transcriptome profiles of expanded versus non-expanded TCR clones looked similar within a particular patient.

We agree with the Reviewer that our scRNAseq analysis consist only of two H7N9 patients, due to the availability of those longitudinal samples. We have stated this in the manuscript and toned down our conclusions.

Results page 9: "It is important to note that our scRNAseq analysis only consist of two H7N9 patients over two time-points, due to the availability of those longitudinal samples."

3. I also have concerns with the mouse experiment and how that would help the understanding of H7N9 human infection apart from the confirmation of T cell activation is associated with flu infection

The mouse data have been removed from the manuscript, as explained to Reviewer 1.

4. The most interesting experiment in this paper is the T cell receptor repertoire analysis which is novel, i.e stable TCR repertoire of A2 M58-66 specific T cells during acute H7N9 infection, however the conclusion of ineffective early clonal expansion is premature and only drawn from limited number of patients (5 survival vs 2 fatal cases) with diverse HLA background are not solid (with two survivals with bearing HLA-A2 with a well know clonal expanded population which in my view should be excluded for analysis)

We agree with the Reviewer that our TCR data represent the most novel findings of our manuscript and such longitudinal TCR analyses during human acute influenza virus infection, for either epitope-specific A2⁺M1₅₈⁺ or activated CD38⁺HLA-DR⁺ CD8⁺ T cells, have not been previously performed. We have expanded our TCRαβ repertoire analysis and solidified our conclusions.

Furthermore, we have compared TCRαβ repertoires within CD8⁺ T cells directed at the single-specificity (A2⁺M1₅₈⁺CD8⁺ T cells) and within activated CD38⁺HLA-DR⁺ CD8⁺ T cells, most likely representing several T cell specificities restricted by different HLA-A and HLA-B alleles. Our data show that TCR repertoire within CD38⁺HLA-DR⁺ CD8⁺ T cells is more diverse at the gene segment usage, both within HLA-A*02:01-expressing donors and non-HLA-A*02:01-expressing donors, than within single specificity CD8⁺ T cells. Thus, we have not excluded HLA-A*02:01-expressing patients from our analyses. However, we have included information on HLA types of our donors used for TCRαβ analysis in our manuscript (Supplementary Table S6).

As explained above, we have toned down our conclusions about early clonal expansions, including the Title, Abstract, Results and Discussion.

Reviewer #1 (Remarks to the Author):

This reviewer thanks the authors for doing an excellent job of revising the manuscript. It now offers a much more interesting, focused, and coherent story - one that should be of interest to the readers of Nature Communications. I have no further comments.

Reviewer #2 (Remarks to the Author):

The revised manuscript has responded to main concerns I raised in great lengths, and it is satisfactory, I have no further concerns.